# iDirac: a field-portable instrument for long-term autonomous measurements of isoprene and selected VOCs

Conor G. Bolas[1], Valerio Ferracci[2], Andrew D. Robinson[3], Mohamed I. Mead[2], Mohd Shahrul Mohd Nadzir,[4] John A. Pyle[1,5], Roderic L. Jones[1] and Neil R.P. Harris[1,2]

[1]Department of Chemistry, University of Cambridge, Lensfield Road, Cambridge, CB2 1EW, UK
[2]Centre for Environmental and Agricultural Informatics, Cranfield University, College Road, Cranfield, MK43 0AL, UK
[3]Schlumberger Cambridge Research, Madingley Rd, Cambridge, CB3 0EL, UK
[4]School of Environmental and Natural Resource Sciences, Universiti Kebangsaan Malaysia, 43600, Bangi, Selangor, Malaysia
[5]National Centre for Atmospheric Science, NCAS, UK

*Correspondence to*: Neil R.P. Harris (neil.harris@cranfield.ac.uk)

## Abstract

The iDirac is a new instrument to measure selected hydrocarbons in the remote atmosphere. A robust design is central to its specifications, with portability, power efficiency, low gas consumption and autonomy as the other driving factors in the instrument development. The iDirac is a dual-column isothermal oven gas chromatograph with photoionisation detection (GC-PID). The instrument is designed and built in-house. It features a modular design, with novel use of open-source technology for accurate instrument control. Currently configured to measure biogenic isoprene, the system is suitable for a range of compounds. For isoprene measurements in the field, the instrument precision (relative standard deviation) is ±10%, with a limit of detection down to 38 pmol mol$^{-1}$ (or ppt). The instrument was first tested in the field in 2015 in a ground-based campaign, and has since shown itself suitable for deployment in a variety of environments and platforms. This paper describes the instrument design, operation and performance based on laboratory tests in a controlled environment, and during deployments in forests in Malaysian Borneo and in Central England.

## 1 Introduction

Isoprene ($C_5H_8$) is one of the most important non-methane biogenic volatile organic compounds (BVOC) emitted into the atmosphere. It has a global emission rate estimated at around 500 TgC year$^{-1}$ (Guenther et al., 2006) and its oxidation products make it a major factor determining ozone and secondary organic aerosol production. Emitted by vegetation, it has been linked to temperature regulation, reducing drought-induced stress and other physiological processes within plants (Sharkey, 2013; Sharkey et al., 2008). A dialkene, isoprene is prone to oxidation by reaction with the hydroxyl radical (OH), as well as by ozonolysis and reaction with the nitrate radical ($NO_3$) (Stone et al., 2011). Isoprene oxidation pathways are complex (Archibald et al., 2010) and result not only in a number of oxygenated volatile organic compounds (OVOCs *e.g.*, formaldehyde, methacrolein, methyl vinyl ketone) but also in a suite of low-volatility stable products and intermediates that can act as precursors of secondary organic aerosols (Carlton et al., 2009; Claeys, 2004; Liu et al., 2016). As a result of its high reactivity and large emissions, determining the global abundance of isoprene is important to understand the oxidising capacity of the atmosphere (Squire et al., 2015) and the formation of SOA, which can affect the optical properties of the atmosphere and in turn impact the climate (Carslaw et al., 2010).

Due to its high reactivity, isoprene is relatively short-lived, with a typical lifetime of the order of one hour in a temperate forest (Helmig et al., 1998). Isoprene emissions are mainly driven by incoming solar radiation and temperature, and as a result exhibit a distinctive diurnal cycle which peaks around midday. Local abundances can change rapidly in response to meteorological variations, such as changes in incoming photosynthetically active radiation (PAR), temperature and atmospheric dynamics (Langford et al., 2010). High time resolution data is required to capture trends in isoprene concentrations

in real time. It is expected that isoprene emissions will be affected by global change (increasing temperatures, land use change, increasing $CO_2$) in the coming decades (Bauwens et al., 2018; Hantson et al., 2017; Squire et al., 2015). However, the overall magnitude and sign of changes in isoprene emissions is still uncertain due to the many variables at play and the uncertainties in our emission models. This, coupled with its large variability, makes it highly desirable to improve the temporal and spatial coverage of isoprene measurements so that our understanding of its emissions via models can be validated against field data.

Measurements of atmospheric hydrocarbons such as isoprene are challenged by the difficulty in making measurements in remote places. To date, in-situ measurements of isoprene have been carried out using existing commercial bench-top instruments, such as gas chromatographs (Jones et al., 2011) and mass spectrometers (Noelscher et al., 2016; Yáñez-Serrano et al., 2015). These techniques differentiate between VOCs either by separation (gas chromatography) or by identification of their molecular ions based on mass-to-charge ratios (mass spectrometry). These instruments, while offering high precision and stability, are not built to withstand field conditions for long periods of time due to their need for power, temperature-controlled environments and speciality carrier gases. This is especially true in under-sampled regions of high isoprene emissions, which are typically in remote or challenging environments (e.g., tropical forests). In these locations instrument size, portability, autonomy, power demand and gas consumption severely limit the length of a deployment. In addition, instrument cost and maintenance limit the number of instruments deployed at any one time, and hence the spatial coverage of a field campaign.

An alternative method to detect environmental VOCs is with grab samples (Robinson et al., 2005). These can either be whole air samples or adsorbent tubes, where air samples (or some specific air components) are collected in an inert vessel and analysed at a later date. While grab samples can be deployed in relatively large numbers, they typically provide low temporal resolution, making this approach unsuitable to capture the rapidly changing concentrations of isoprene. In addition, reactive compounds can degrade over time before analysis, and using this method for long periods, even with some degree of automation, is very time and resource intensive. Recent work showed that it is possible to retrieve isoprene abundances in the boundary layer using satellite measurements by means of thermal infra-red imaging (Fu et al., 2019). However, with uncertainties in the range of 10-50%, these retrievals would benefit from further validation from ground-based instrumentation.

All the limitations in the instruments currently used for VOC detection drive the need for a field instrument that is:
- lightweight, so that it is portable and can easily be carried and installed in environments difficult to access with traditional instrumentation;
- low-power, so that it is capable of running off-grid, allowing measurements in locations with no mains power;
- autonomous, so that it minimises operator involvement and maintenance;
- low gas use, so that it minimises the cylinder size required and the number of site visits to replace gas cylinders;
- rugged and durable, so that it can withstand challenging environments; and
- relatively low-cost, so that many instruments can be deployed at one time, maximising spatial coverage.

Here we describe the development and validation of the iDirac, an instrument that fulfils the requirements listed above. It follows on from the philosophy of the $\mu$Dirac (Gostlow et al., 2010), with portability, modularity, power efficiency, and autonomy at the centre of its design. The iDirac also incorporates inexpensive microcontroller board processors for advanced control and remote access to the instrument. The core GC instrument and its operation are described in Section 2, while Section 3 presents the software used to control the instrument. Instrument performance is discussed in Section 4, including calibration, accuracy, precision, sensitivity and separation ability. Finally, results from trial runs in the controlled environment of a laboratory and deployments in Malaysian Borneo and Central England are presented in Section 5. Results have been published

on the impact of herbivory on canopy photosynthesis and isoprene emissions in a UK woodland (Visakorpi et al., 2018) and on isoprene concentrations near the Antarctic peninsula (Nadzir et al., 2019).

## 2 Practical description of the iDirac

The iDirac is a portable gas chromatograph equipped with a photoionisation detector (GC-PID): the VOCs in an air sample are separated on chromatographic columns and then sequentially detected by the PID. The instrument is built in-house and is lightweight, low-power and able to operate for up to several weeks or months autonomously. Its specifications are shown in Table 1. Section 2.1 describes the basic outline of the instrument and Section 2.2 describes the specific configuration of the instrument for isoprene.

**Table 1: iDirac specifications**

| Power | 12 W |
|---|---|
| Weight | 10 kg |
| Voltage Requirements | 10–18 V |
| Dimensions | $22 \times 61.6 \times 49.3$ cm |
| Carrier Gas | High Purity Nitrogen (Grade 5.0, or 99.999%) |
| Calibration Gas | 10 nmol mol$^{-1}$ (or ppb) high-accuracy isoprene in nitrogen |
| Limit of detection | 38 pmol mol$^{-1}$ (or ppt) |
| Precision | 11 % |

### 2.1 Core gas chromatograph physical design

The iDirac is built in a modular fashion, so that the various components are housed in 6 main plastic boxes (Piccolo Polycarbonate Enclosures, IP67) packed in foam inside a protective waterproof case (Peli 1600), as shown in Figure 1. Details on the boxes and their contents are given below, and shown within the instrument in Figure 1:

- Valve Box, containing 8 solenoid valves to control gas flow from the four inlets;
- Control Box, containing microcontroller boards (Arduino and Raspberry Pi), a number of electronic components (e.g.. solid state relays), the flowmeter and SD card for data storage;
- Oven Box, containing the dual-column system, (pre- and main columns), heating element and Valco® valve;
- PID Box, containing the photoionisation detector (PID);
- Pump Box, containing the pump and pressure differential sensor;
- Power Box, contains power regulators and electrical fuses.

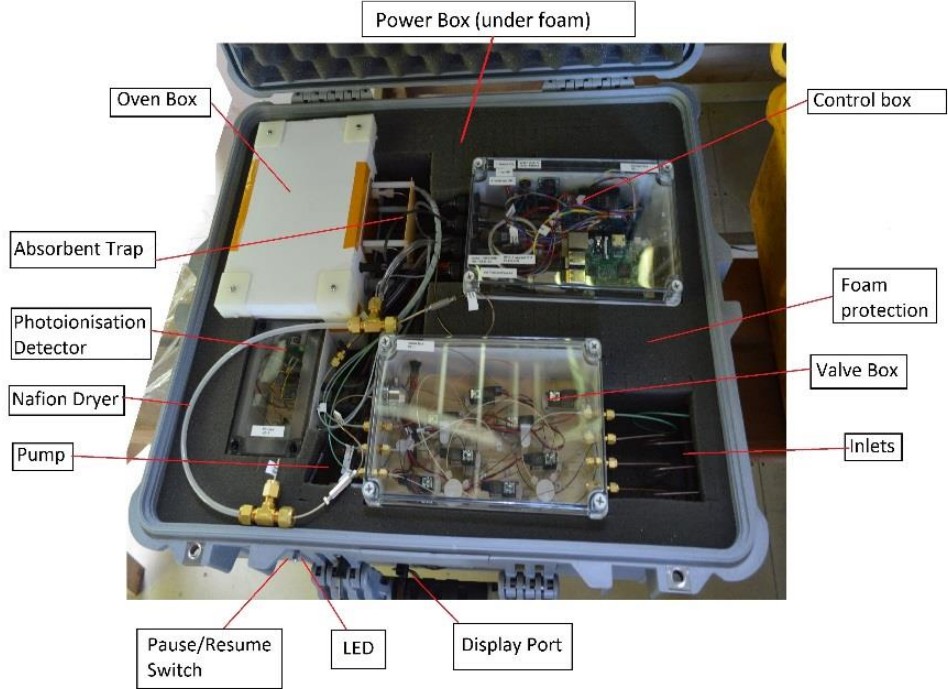

**Figure 1: Interior of the iDirac showing the modular design of its component parts inside the main Peli case (22 × 61.6 × 49.3 cm).**

On the exterior, the iDirac has a power socket, and four inlets for gas input. The inlets are for the nitrogen carrier gas, a calibration gas and two sample lines (Sample 1 and Sample 2) between which the instrument can alternate.

The general pneumatic design of the instrument is built around two phases in the analysis cycle which are represented schematically in Figure 2: a loading phase (Load Mode – pink), in which the analyte of interest is pre-concentrated on an adsorbent trap, and an injection phase (Inject Mode - purple), in which the analyte is desorbed from the trap and directed into the oven for separation and, eventually, detection. These two modes are controlled by a 2-way 10-port Valco® valve (VIDV22-3110, mini diaphragm 10 port 2-pos 1/16" 0.75mm, Thames Valco) in the Oven Box, which is activated by pneumatic actuation, by the set of solenoid valves in the Valve Box, and by the pump.

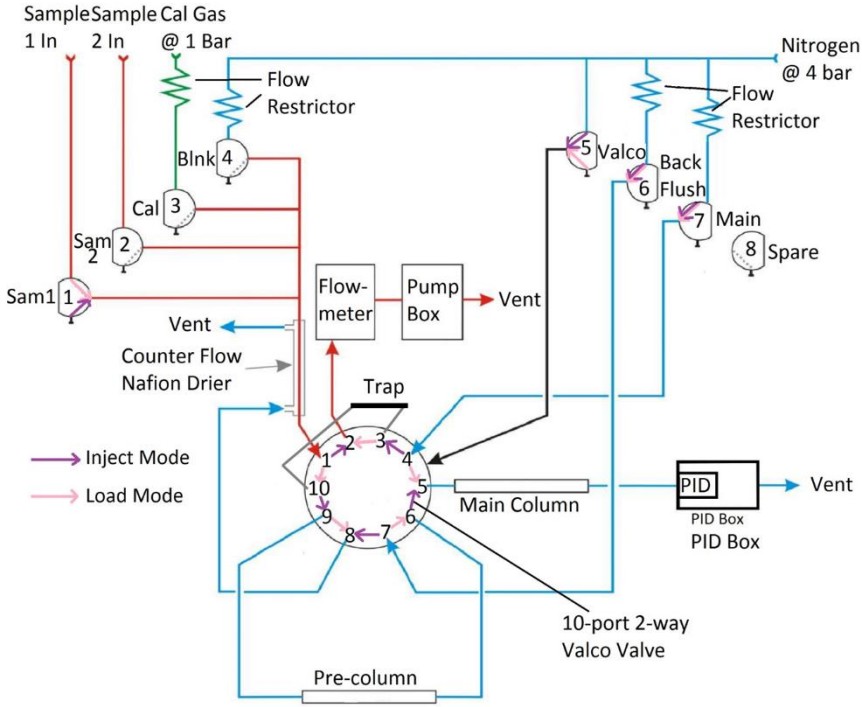

**Figure 2: Schematic representation of the iDirac operation. When in Load Mode (valve 5 off - pink), the contents of a gas source chosen between valves 1-4 are pre-concentrated on the adsorbent trap. In Inject Mode (valve 5 on - purple), the VOCs in the trap are injected into the dual-column system for separation and, eventually, detection.**

In Load Mode (Valco valve not activated, i.e. valve 5 off), one of four inlet gases (either sample 1, sample 2, calibration gas or blank gas) is selected by switching on the appropriate solenoid valve (valves 1, 2, 3, or 4 respectively). By activating the pump, gas is drawn through the selected inlet valve, dried in a Nafion counter-flow system and passed through an adsorbent

trap where the analyte is pre-concentrated.  The sampled gas is vented into the Peli® case and then to the outside. A flowmeter is placed in series with the sample flow and measures the gas flow through the trap. Once a pre-defined volume of gas has been sampled, the pump stops and the instrument enters Inject Mode. Laboratory tests found no statistically significant difference in isoprene peak area between runs using the drier and runs bypassing it.

In Inject Mode, the trap is flash-heated to $300 \pm 5$ °C for 9 s to desorb the analyte from the adsorbent material. The Valco® valve is then pneumatically activated by switching valve 5 on: the nitrogen carrier is flowed through the trap in the direction opposite to trap-loading, delivering the desorbed compounds into the dual-column system where they undergo chromatographic separation. The oven consists of a pre-column, which screens for large molecules (e.g., the monoterpenes) whilst allowing smaller molecules through, and a main column, which performs the critical separation of the relevant analytes.

The main column eluent is incident on the PID membrane, where the signal from the changing composition of the gas exiting the main column is detected.

More details on the individual parts of this cycle are given below.

**Inlet manifold and sample preparation.** The inlet ports protrude from the side of the Peli® case via 1/16" inch bulkhead unions (Swagelok) and connect directly to the Valve Box, containing 8 solenoid valves that act as gas selectors. The Sample

1 (via valve 1), Sample 2 (via valve 2), calibration gas (via valve 3) and blank nitrogen (via valve 4) lines are all combined in a 4-way Silconert-treated stainless steel Valco manifold (Z4M1, 1/16" manifold 4 inlets, Thames Valco). This manifold leads to the adsorbent trap via a Nafion dryer (Nafion gas dryer 12", polypropylene, PermaPure MD-050-12P-2) which drives excess water vapour out of the gas flow by diffusion through a membrane with a counter flow of dry high-purity nitrogen. Valve 5 is a direct line from the nitrogen inlet to the Valco valve for actuation, which requires a higher pressure (typically 4 bar). Valves

6 and 7 control the nitrogen flow through the columns: valve 7 activates the nitrogen flow through both columns in Inject Mode (when valve 5 is on), and through the main column only in Load Mode (when valve 5 is off). Valve 6 activates the nitrogen flow through the pre-column for the backflush in Load Mode. The nitrogen counter-flow needed for the Nafion dryer is provided by Valve 6 in Inject Mode and by the pre-column backflush vent in Load Mode. Gas lines downstream from valves 5, 6 and 7 leave the box via manifolds on the side of the box. Valve 8 is a spare valve with no current function.

Flow restrictors upstream from valves 3, 4, 6 and 7 ensure that the flow from the pressurised inlet lines does not exceed the maximum flow through the flowmeter, and also reduce the gas demand of the instrument. The restrictor tubing used for the calibration line is red PEEK flow restrictor (1/32" OD, 0.005" ID) and the one used for the nitrogen lines is black PEEK (1/32" OD, 0.0035" ID). The rest of the tubing is Silconert-treated stainless steel (Thames Restek, 1/16" OD, 0.04" ID), which does not restrict the gas flow.

**Sample adsorption/desorption system**. From the Nafion drier, the sample gas passes through ports 1 and 10 of the Valco valve and into the adsorbent trap when the instrument is in Load Mode. The trap consists of wide bore stainless-steel tubing (HI-Chrom, 1/16" OD, 0.046" ID) containing one bed of adsorbent material between two beds of glass beads, both crimped in place, with a coiled nichrome wire heating element surrounding the section of the tube corresponding to the adsorbent. The nichrome wire has a ceramic electrically insulating coating to prevent shorting with the trap tubing. The adsorption of isoprene

and other VOCs takes place on a bed of approximately 10 mg Graphsphere 2016 (formerly Carboxen 1016, Supelco, 60/80 mesh, 11021-U); Graphsphere 2016 is a carbon molecular sieve that has been selected for its optimised recovery rate of unsaturated short chain hydrocarbons upon thermal desorption. Different sorbent materials can be used for other species. The gas exiting the trap, now scrubbed of VOCs, flows via ports 3 and 2 on the Valco valve into the flowmeter (Sensirion,

ASF1430) which monitors the flow rate through the trap. This is then integrated across the duration of sampling to calculate the total volume of gas sampled. When the desired volume (as specified by the user in the configuration step – see Section 3) is reached, the valves from the sample inlet are closed and the pump is halted to stop the flow of gas through the trap. The heating coil is flash-heated to desorb the analyte from the adsorbent, while the Valco valve is switched to Inject Mode and

valve 7 is activated, flushing the desorbed VOCs onto the pre-column in the oven box with the high-purity nitrogen carrier.

**Isothermal oven.** The flow containing the sample leaves the trap and enters the thermally insulated oven box. This enclosure, housed in insulating material (Lightweight display board, Kerbury Group), is heated to 40 $^o$C using a heating element (PTC element enclosure heater, 15W 12-24V 40C) which is fixed to the base-plate of the oven using conductive paste. A fan mixes the air inside the oven to ensure a uniform temperature throughout. The oven temperature exhibits diurnal variations (typically

in the range of $\pm 2$ $^o$C) that appear driven by ambient temperature. This introduces some variability in the isoprene retention time, but it is accounted for in the analysis of chromatograms (see Section 3.3).

The sample is injected onto the pre-column (5% RT-1200, 1.75% Bentone-34, SILPT-W, 100/120, 1.0 mm ID, 1/16"OD SILCO NOC, Custom Packed, Thames Restek, ~70 cm in length) via ports 10 and 9 on the Valco valve. Here, isoprene and other small molecules travel faster through the pre-column than bulky VOCs. After a set time (typically, 30 s), once isoprene

has passed through the pre-column, the Valco valve is switched off, with Valve 5 closing and Valve 6 opening, so that the pre-column is back-flushed. This way lighter species, including isoprene, elute onto the main column while larger molecules that are still in the pre-column when valve 5 is switched off are removed from the column system via the back-flush.  This is important to avoid large, less volatile species from entering the main column.

The main chromatographic separation occurs on the main column (OPN-RESL-C, 80/100, 1 mm ID, 1/16"OD, SILCO NOC,

Custom Packed, Thames Restek, ~70 cm in length), based on the boiling point and polarity of the VOCs.  This way, different species elute onto the detector at different times.

**Photoionisation detection system.** The sample is directed from the outlet of the main column into a photoionisation detector (PID). The PID (Alphasense Ltd™, PID-AH) operates by ionising any gas diffusing through a membrane covering a krypton lamp. Near-vacuum UV radiation from the lamp ionises any molecule with an ionisation potential of less than or equal to 10.6

25   eV. Isoprene, with an ionisation potential of 8.85 eV (Bieri et al., 1977), is readily photolysed and hence detected by the PID with a sensitivity of 140% relative to that of isobutylene, which is used by the manufacturer as a reference compound in terms of PID response. The ions generated by photoionisation produce a voltage change across an electrode system which is converted to a digital signal by an analogue to digital converter (ADC) (16-Bit ADC 4 Channel, Adafruit). The PID is turned on for the duration of the elution from the dual-column system, and the data is collected at a frequency of 5 Hz.  The

chromatography run finishes once isoprene has eluted from the main column (typically 60-75 s after starting the back-flush). The data from the PID is then saved to a new file on an SD card by the Arduino Mega. A typical chromatogram showing an isoprene peak is shown in Figure 3.

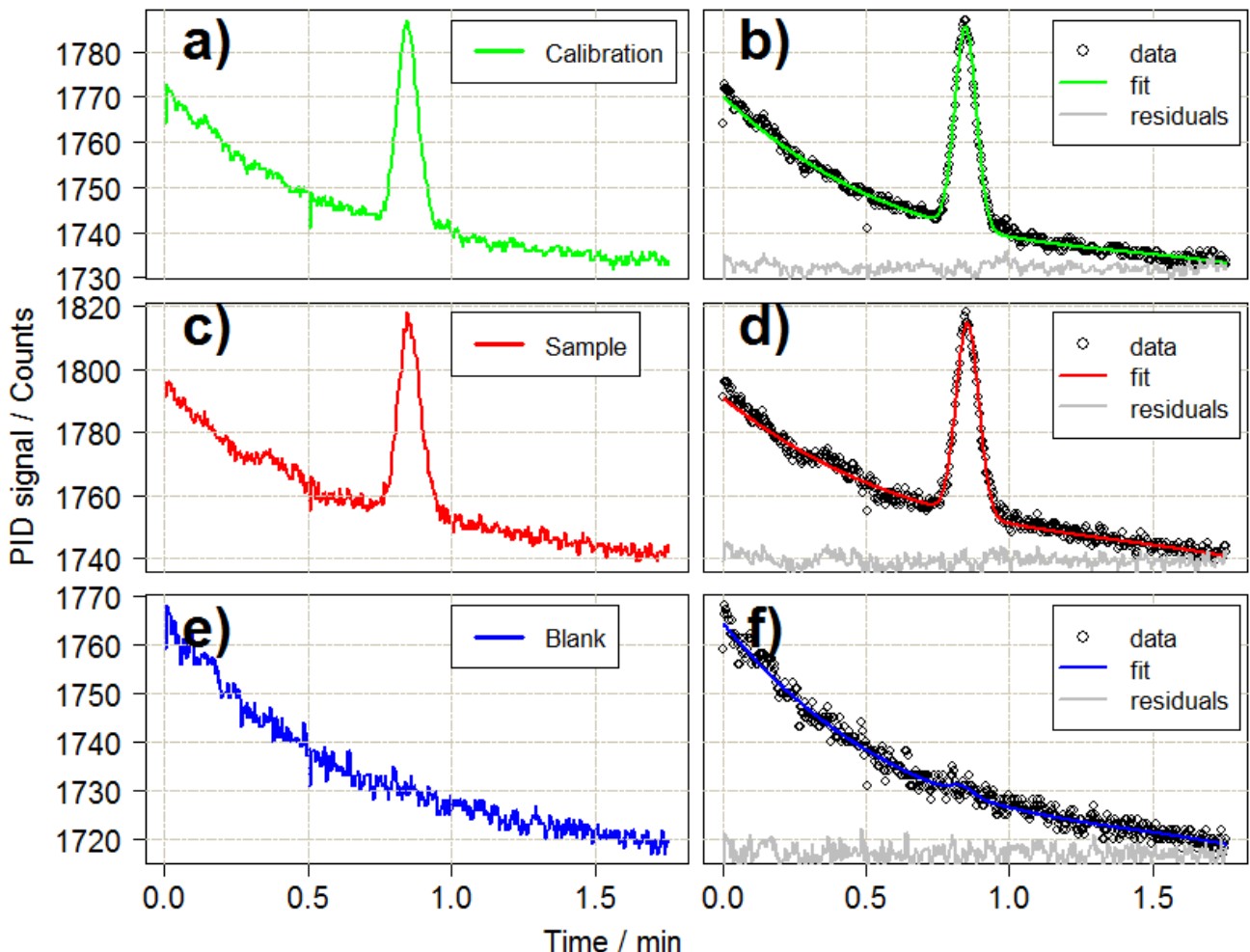

**Figure 3: Typical chromatograms for a) calibration, c) sample, and e) blank runs. The isoprene peak detected by the PID is around the 0.8 min mark. Panels b), d) and f) show the combined baseline and Gaussian fits to the observed data for each run type. Residuals are offset for clarity. All the chromatograms are from the deployment in Wytham Woods (see Section 5.3). The calibration run is for 12 mL of a 11.6 ppb standard of isoprene in a nitrogen balance. The sample run is for a 150 mL ambient air sample (later quantified as 1.5 ppb isoprene). The blank run is for a 12 mL sample of Grade 5.0 nitrogen.**

### 2.2 Instrument operation specifications

**Carrier gas and calibration gas.** Two gas cylinders are required to operate the iDirac: a pure nitrogen supply and a calibration gas. Nitrogen is used as carrier gas through the dual column system, as sample gas for the blank runs and also to actuate the Valco valve. The nitrogen supply is of at least Grade 5 purity (corresponding to ≥ 99.999 % nitrogen) to minimise interference from impurities with the detection of isoprene. Typically, we use high purity BIP+ Nitrogen (Air Products). The logistics of the measurement dictate the size of the nitrogen cylinder used: for mobile deployments in the field, small portable cylinders (1.2 L) are ideal, whilst larger cylinders (10 L) are more suitable for long-term measurement as they minimise the need for maintenance visits to replace the nitrogen cylinder. Typically, the iDirac can run continuously on a 10 L nitrogen cylinder supplied at 200 bar for approximately 2 months. The calibration gas consists of a binary gas mixture of approximately 10 nmol mol$^{-1}$ (or ppb) isoprene in a nitrogen balance stored in a Silconert-treated 500 mL stainless steel cylinder (Sample Cylinder Sulfinert, TPED, 1/4", Thames Restek). The use of cylinders with passivated internal walls minimises the adsorption of isoprene on surfaces, which would introduce biases in the measurement. The accurate concentration of the calibration gas is determined by comparison with a primary gas standard. The calibration routine is described in detail in Section 4.1.

**Power requirements for operation.** The instrument requires a power supply between 9 and 18 V. This can either be mains power, or alternatively, a battery. The incoming power is smoothed and regulated with two regulators to stable 5 V and 12 V outlets. The Arduino board monitors the supplied voltage in between runs in the case of the battery losing charge or power cuts. If the voltage drops, the iDirac switches to a power-save mode, where the oven, PID and valves are turned off to conserve

power and the instrument waits for 20 minutes before again checking the input voltage. Once a high enough voltage (typically 9 V) is detected, the various components are turned on again.

**Flow control through the instrument.** The flow through the instrument is driven by either upstream pressure (in the case of the nitrogen and calibration gas flows) or by the pump box (in the case of Samples 1 and 2). The pump box is an air-tight container with an inlet line and a vent. A diaphragm pump (DF-18, Boxer) withdraws air from the pump box and vents it outside, reducing the pressure inside the enclosure. The reduced pressure within the pump box causes air (from the Sample 1 and 2 inlets) to be drawn through the system, via the trap and the flowmeter. A pressure sensor (differential pressure sensor, Phidgets) monitors the pressure differential between the inside and the outside of the pump box. During a pump cycle, the pump is only activated when the pressure differential falls below a pre-specified value (nominally 20 kPa). This method ensures a uniform flowrate and enables control over low flowrates ($\sim$ 20 mL min$^{-1}$), thus reducing the uncertainty in the volume integration of the air sampled.

## 3 iDirac software and hardware control and data analysis

The iDirac is controlled using a dual Arduino system: an Arduino Micro board controls the gas flow components of the instrument, whilst the main instrument control is achieved with an Arduino Mega board. These two units communicate with all of the sensors inside the instrument and read their outputs. A Raspberry Pi computer acts as the interface between the user and the Arduino boards. A Python script is run on the Raspberry Pi, allowing the user to configure the instrument with the desired parameters and read the sensor output from that of the Arduino. The Raspberry Pi desktop can be accessed remotely via an ad-hoc network, allowing connection with a variety of interfaces. This control system allows many of the parameters described above (e.g., sample volume, time spent in each column) to be changed.

### 3.1 Arduino control of internal electronics

The instrument is controlled primarily using an Arduino Mega 2560 board (Arduino Mega 2560, Arduino). This microcontroller has a number of analogue and digital ports and runs Arduino code (C and C++ commands) to control these ports. An SD breakout board is used (microSD Card Breakout Board, Adafruit ) to facilitate the use of an SD card to store data in, while a real time clock (RTC) board is used (Real Time Clock, ChronoDot Ultra-Precise, Adafruit) for time-keeping. Figure 4 illustrates the various connections on the Arduino Mega.

An Arduino Micro board (Arduino Micro, Arduino) reads specifically the altimeter pressure sensor (located in the PID box) and the flowmeter, and sends these readings to the Arduino Mega via a serial port. The use of the Arduino Micro is justified as it simplifies the code on the Arduino Mega, particularly as the flowmeter requires the use of a shifter to convert the RS232 serial signal and several subsequent mathematical manipulations. The Arduino boards do not have a shutdown procedure and can simply be unplugged.

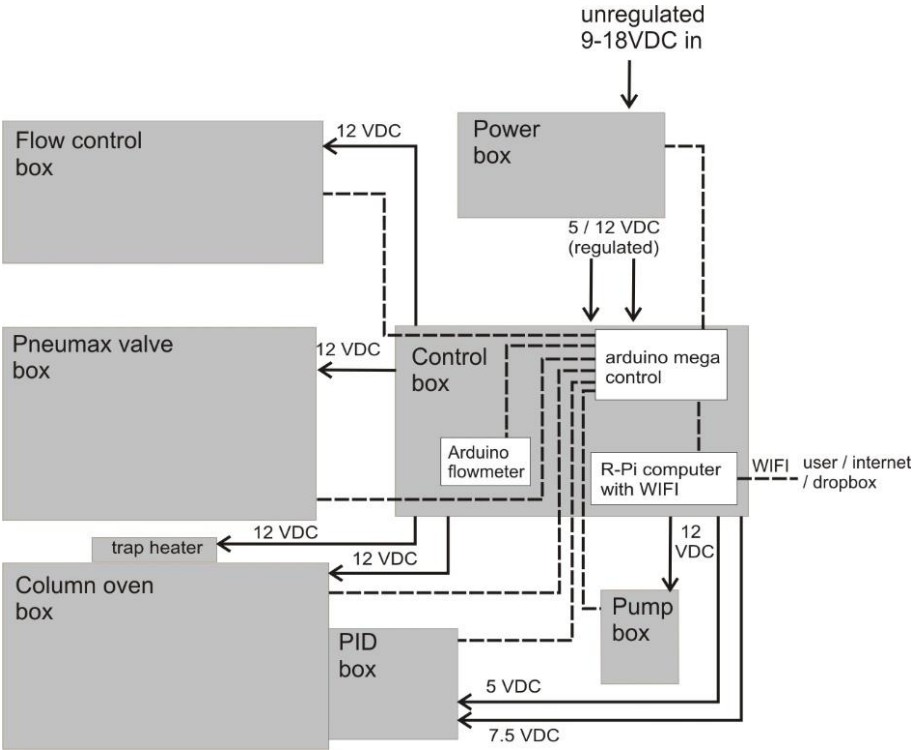

**Figure 4: Schematic of Arduino Mega connections**

### 3.2 Description of Raspberry Pi user interface

The iDirac uses a Raspberry Pi (Raspberry Pi Model B V1.1, Raspberry Pi) as a user interface, allowing the instrument to be controlled from a familiar desktop environment. The Raspberry Pi uses a Wi-Fi dongle to set up its own ad-hoc network, which can be connected to by laptops and mobile phones in a fashion similar to a standard Wi-Fi network. Once connected to the network, a graphical desktop sharing system such as VNC viewer (VNC Viewer, RealVNC) allows the user to navigate the Raspberry Pi desktop and manipulate the instrument.

Upon opening the Raspberry Pi desktop a purpose-written Python script is launched automatically. A terminal window is opened displaying the serial output from the Arduino Mega and transmitting data to the Arduino Mega via a serial port connection. The latest version of this Python script is freely available (https://github.com/cgb36/iDirac-scripts). The Python script decodes incoming serial bytes from the Arduino Mega and displays them in a user friendly command line window. It is also possible to restart and shutdown the instrument from the Raspberry Pi desktop. The Raspberry Pi requires a shutdown procedure, which can be done either physically with a switch on the side of the control box, or from the virtual desktop environment.

### 3.3 Processing of chromatograms

To process numerous chromatograms in an automated fashion, a script was created that uses calibration runs to accurately identify isoprene peaks in the sample runs and convert their integrated peak areas into mixing ratios. This script is written in Mathematica (v11.1.1). Figure 5 shows a flow diagram for the main algorithms of the script. Firstly the data is read in, making sure that all the files are the correct size and do not contain any erroneous runs (e.g., corrupted or truncated files) that may jeopardise the running of the script. Each chromatogram file has an index field, either S, X, C or B which indicate if the chromatogram is a sample 1, sample 2, calibration or blank chromatogram respectively.

The calibration data is processed first. This involves selecting all calibration chromatograms (those with index 'C') and plotting them for visual inspection. From the plot of all calibration chromatograms, the user specifies the regions that are used to fit to the isoprene peak and the baseline. A third-degree polynomial is fitted to the baseline over the user-specified baseline intervals. A Gaussian curve is then fitted to the baseline-subtracted chromatogram over the user-specified peak interval. The peak height,

width and position (equivalent to elution time) of the fitted Gaussian, as well as the error in the fit (root mean square error, RMSE), are logged. The elution time of the peak is retained in an interpolated function over time. This function is then used to locate the isoprene peak in the sample runs between two calibration runs. The blank runs (with index 'B') are included in this routine as they effectively represent calibrations with zero isoprene concentration. Subsequently, the peak area is plotted against the number of calibrant moles to obtain a response curve. The number of calibrant moles, $n_{cal}$, is defined as:

$$n_{cal} = (V_{cal}/V_{mol}) \cdot \chi_{cal} \qquad\qquad (1)$$

where $V_{cal}$ is the sampled volume of the standard during the run, $V_{mol}$ is the molar volume of an ideal gas, and $\chi_{cal}$ is the isoprene amount fraction in the gas standard. A straight line is then fitted to this data. Calibration procedures are described in depth in Section 4.1.

The sample chromatograms are then selected as either Sample 1 (runs with index 'S'), or Sample 2 (runs with index 'X') and, as with the calibration runs, they are plotted to visually inspect the data. Following that, we interpolate the retention times from adjacent calibration runs to the time of each sample runs, thus ensuring that the isoprene peak is identified correctly. This effectively takes into account variation in elution time caused by varying oven temperatures. A baseline is fitted to the sample chromatograms in a similar fashion to those fitted to the calibration ones. Then a Gaussian function, constrained by certain boundaries (e.g., peak width within the average calibration peak width ± 1 standard deviation, retention time within ±4 s of the interpolated retention time), is fitted to the section of the chromatogram indicated by the interpolated calibration retention times. Using the sample peak area ($A_{sam}$), the sample volume ($V_{sam}$) and the intercept ($c$) and gradient ($m$) of the calibration curve, the isoprene mixing ratio in the sample, $\chi_{sam}$, is calculated using Eq. (2):

$$\chi_{sam} = (A_{sam} - c)/m \cdot (V_{mol}/V_{sam}) \qquad\qquad (2)$$

When there are insufficient calibration chromatograms to determine the isoprene peak retention time (e.g., less than 4 calibration runs in a day), it can be estimated using the column temperatures from the nearest calibration runs. If the spacing between calibration points is too great or the calibration is done separately to the sampling, the interpolated calibration retention time values may not span the region where the isoprene peak resides. In this case the column temperature and retention time of the most recent calibration chromatograms are used to define a linear relationship. It is then possible to derive the isoprene retention time from the column temperature of the sample chromatogram.

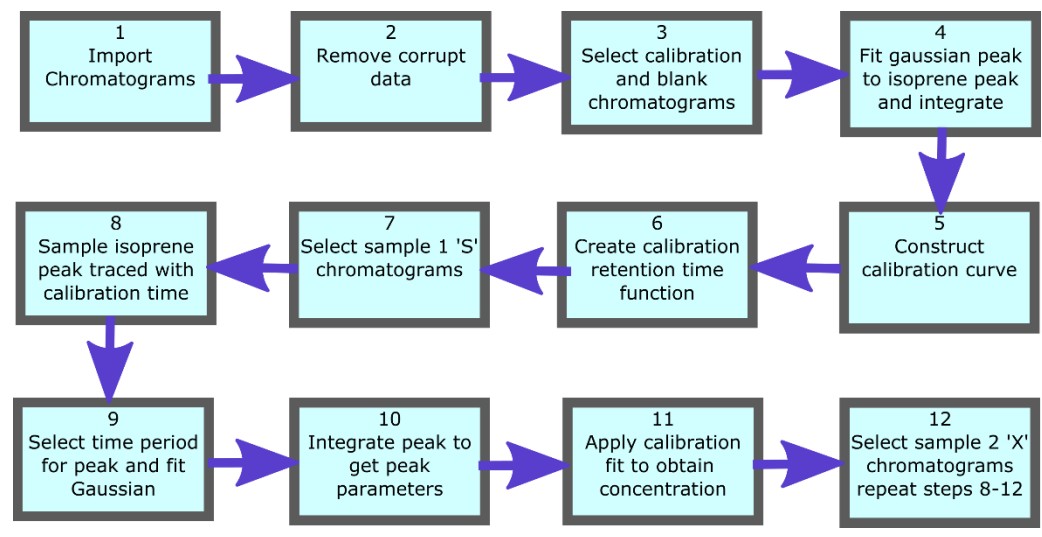

**Figure 5**: **Analysis script flow diagram**

## 4 Instrument performance

### 4.1 Calibration of output chromatograms

The PID response to isoprene is calibrated using a primary gas standard supplied by the National Physical Laboratory (NPL), certified as containing $5.01 \pm 0.25$ nmol mol$^{-1}$ (or ppb) isoprene in a nitrogen matrix (uncertainty provided at the $k = 2$ level). The gas mixture is stored in a 10 L Experis cylinder (Air Products); this type of cylinder has been demonstrated to provide maximum stability (up to 2 years) for VOC mixtures over time (Allen et al., 2018; Rhoderick et al., 2019). The primary standard is only used for calibration in the laboratory; for field deployments, a smaller secondary gas standard is used instead. This is prepared manometrically by diluting a higher concentration parent mixture (100 nmol mol$^{-1}$ isoprene in nitrogen, BOC) to approximately 10 nmol mol$^{-1}$ with high-purity nitrogen (BIP+, Air Products). This secondary gas standard is prepared in a 500 mL Silconert-treated stainless steel cylinder (Sample Cylinder Sulfinert, TPED, 1/4", Thames Restek). This type of treated cylinder exhibits very good long-term stability for a number of VOCs (Allen et al., 2018; Rhoderick et al., 2019). The exact isoprene amount fraction in the secondary standard is determined by validating it against the NPL primary standard. This way the measurements from the iDirac are traceable to accurate primary standards. We routinely measure the secondary standards against the primary standard before and after field deployments to account for any degradation over time. However we have found no statistically significant degradation over the time span field deployments (up to 5 months).

Frequent calibration is needed not only to convert chromatogram peaks into mixing ratios, but also to monitor long-term trends in the detection system, including detector drift and decreasing performance of the adsorbent trap. Any changes in isoprene elution time, which may be caused by changes in oven temperature, can affect the correct peak assignment in chromatograms with multiple peaks. These effects can be easily addressed if frequent calibration chromatograms (which only have, by definition, one peak) are available.

Calibration frequency is specified by the user in the instrument set-up by selecting the number of samples to run between calibrations. For example, a calibration frequency of '4' would correspond to a run of four sample chromatograms, followed by a calibration run. It is essential to perform a calibration run periodically to ensure that the position of the isoprene peak can be traced. Typically, a calibration run is performed every 35 sample runs. As the mean duration of a 150 mL sample run is approximately 9 min (consisting of 7.5 min of sampling and 1.5 min of chromatographic run), a calibration run is performed approximately every 5.25 hours.

The calibration cycle is programmed to be preceded and followed by a blank run, in which the system samples from the high-purity nitrogen supply from Valve 4. This allows any residual isoprene in the trap to be desorbed before and after calibration, and to monitor the efficiency of desorption over time.

A calibration curve is obtained by varying the volume sampled in each calibration run. When configuring the instrument, the user specifies a calibration volume in mL, which is sampled every other calibration run. For the remaining calibration runs, the instrument is programmed to sample a volume picked randomly from 5 possibilities: the user-specified calibration volume, the user-specified calibration volume multiplied by 2 or 4, and the user-specified calibration volume divided by 2 or 4. For instance, for a run configured with a calibration volume of 12 mL, half the calibration runs would be of 12 mL samples and half a random mixture of 3, 6, 12, 24 and 48 mL samples. A typical time sequence of isoprene peak areas from different calibration volumes is shown in Figure 6. A calibration curve is then obtained by plotting these peak areas against the number of calibrant moles (as defined in Eq.(1)). The zero moles point is obtained from the blank runs. A straight line is fitted to the calibration data. A typical calibration plot is shown in Figure 7. The straight line fit allows the determination of the fractional isoprene amount in the samples via Eq. (2) by extrapolation or interpolation, provided the sample volume and peak area are known. Typically, data is analysed in weekly segments, so that a calibration curve is obtained for each week.

The error in the sampled volumes is dominated by the dead volume in the gas lines before the trap (approximately 1.6 mL), combined with the uncertainty in the measurement of flow rates (1%) and sampling times (0.05%). The overall uncertainty in the volumes is estimated as 50% for 3 mL, 13% for 12 mL, 3% for 48 mL and 1% for 200 mL.

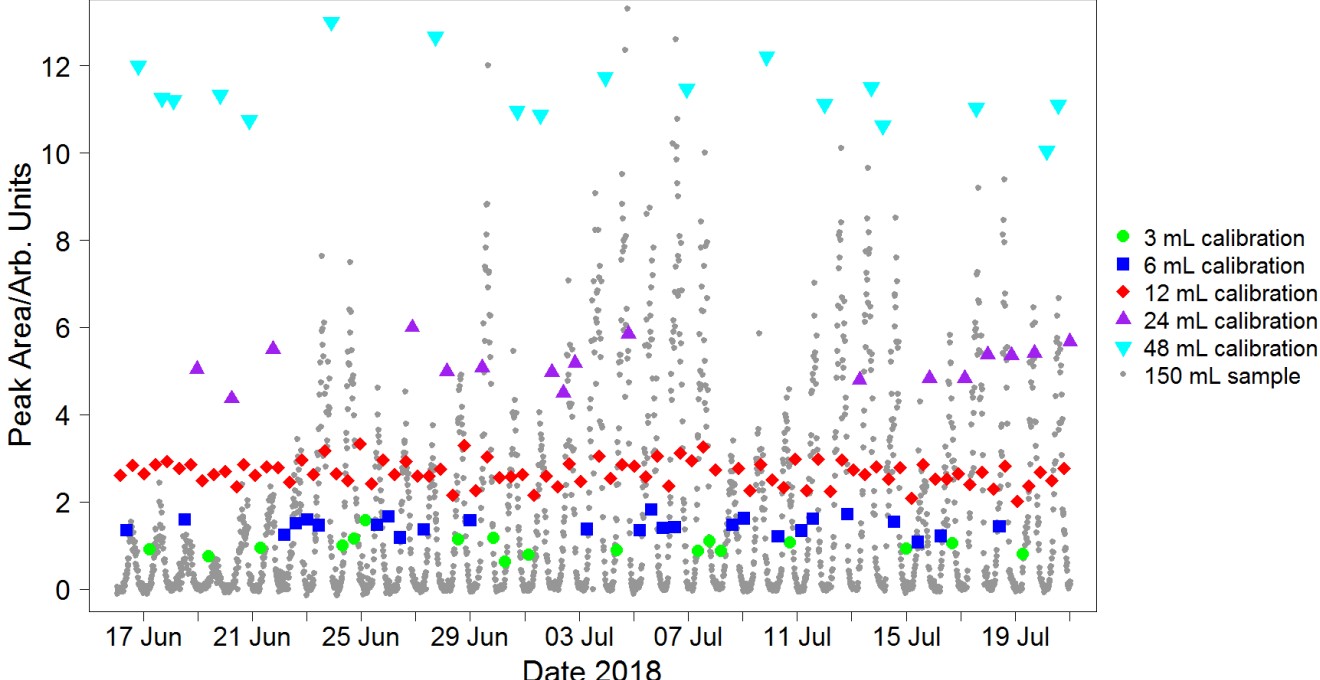

**Figure 6: Typical sequence of isoprene peak areas for runs with varying calibration volumes. These, once split into weekly segments, are used to produce a calibration curve (see Figure 7). The calibration runs with the standard user-specified sampled volume (red data points) are used to calculate the instrument precision on a weekly basis (see Section 4.2). Peak areas from sample runs (grey data points) are also shown to illustrate how the calibration peak areas span the entire range of sample values, minimising the need for extrapolation. This plot was produced using data from the Wytham field campaign (see Section 5.2).**

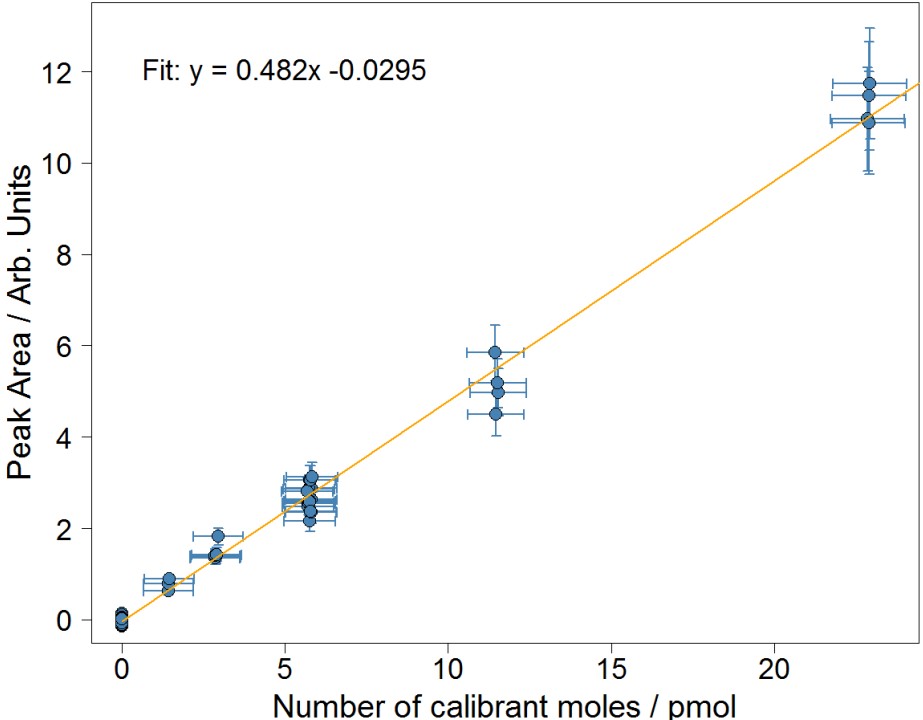

**Figure 7: Calibration plot for isoprene for the week of 02-08/07/2018 from the Wytham field campaign. Error bars in the area correspond to the precision of the measurement (±10.4%). Error bars in the calibrant moles are estimated from the uncertainties in the secondary standard used and in the volume of gas sampled.**

As interpolation carries lower uncertainty than extrapolation, it is important to choose an appropriate value for the user-specified calibration volume, so that the points in the calibration curve span the entire range of the sample runs (as is the case in Figure 6). Typically, 12 mL is suitable in an environment with relatively low (< 1 ppb) isoprene concentrations (e.g. remote oceans), whilst a higher value (20 mL) is more appropriate when measuring in areas such as tropical forests.

## 4.2 Precision and accuracy of iDirac data

**Precision.** The precision of the instrument was determined as the relative standard deviation in isoprene peak area from calibration chromatograms with the same user-specified volume (typically, more than 50% of the total calibration runs in any given measurement sequence, as detailed in Section 4.1) and from the same calibration cylinder. For instance, in the calibration sequence shown in Figure 7, this corresponds to the runs of 12 mL samples. Following analysis of the scatter of these data points, the instrument precision determined as ± 10.4% in the field (compared to <5% in the laboratory). This procedure is carried out for each weekly segment of the data so that the measurement precision can be routinely monitored over time which is especially useful in long deployments.

**Accuracy.** One of the main components of the accuracy of the instrument is the uncertainty in the isoprene amount fraction in the NPL standard, and how this is propagated to the isoprene amount fraction in the secondary gas standard used in the field. It is therefore essential that the concentration of the secondary calibration cylinder is determined as accurately as possible by comparing it to the NPL primary standard. This is carried out in the laboratory, typically before and after each field deployment. An example of this concentration determination is shown in Figure 8. XLGENLINE, a freely available generalised least-squares (GLS) software package for low-degree polynomial fitting (Smith, 2010) is used to estimate the uncertainty in the isoprene amount fraction in the secondary calibration cylinder. This is carried out in two steps. First the calibration data (i.e., the peak areas and sampled volumes from the NPL primary standard) is run through XLGENLINE to produce a calibration line with associated uncertainty envelope. In the second step, this calibration curve is used to convert the peak areas from the secondary standard (i.e., the "unknown") into concentrations and their associated uncertainties. For secondary calibration cylinders, this is estimated as ~ 3.5% (1 standard deviation). A similar procedure is applied to field data to estimate the uncertainty in the ambient isoprene concentrations (now using the secondary standard for the calibration). This is estimated as ~10-12.5 % (1 standard deviation).

Co-elution of interfering species can also affect accuracy. Tests targeting specific potential interferents are described in Section 5.1 and show that these species do not overlap with the isoprene peak in the chromatograms. However co-elution with unknown (or not tested for) species, albeit unlikely, can never be fully ruled out. If these species have longer lifetimes than isoprene, the observed night-time abundances attributed to isoprene can be used as the upper limit of potential interference of unknown co-eluters (assuming they are trapped with the same efficiency and have the same PID response as isoprene). The isoprene night-time mixing ratio is 50 ppt for the data shown in both Figures 14 and 15. Therefore we estimate the instrument accuracy for field data as the combination of the propagated uncertainty from the standard (10-12.5 %) and the potential co-elution of long-lived species (50 ppt). This correspond to an overall accuracy of ± 1.2 ppb for a 10 ppb isoprene sample, ± 0.13 ppb for a 1 ppb isoprene sample and ± 51 ppt for a 100 ppt isoprene sample.

Deviations of peak shape from a simple Gaussian function also impact accuracy by introducing a bias in the reported peak areas. However this is limited to high volume, high concentration samples and can add ~2% to the overall accuracy budget.

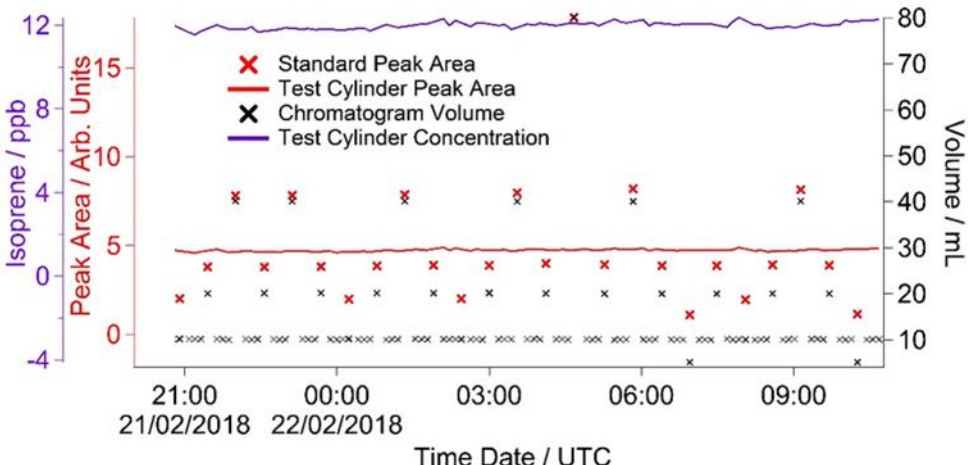

**Figure 8: Summary plot of a concentration determination experiment. The primary reference gas mixture is used as the standard in the calibration runs, and the secondary gas mixture under test is used as sample.**

### 4.3 Sensitivity of the iDirac to isoprene

The instrument's sensitivity can be adjusted by changing the volume of the sample being analysed. For high concentrations (e.g. strong leaf emissions) a smaller volume should be used a smaller volume should be used to avoid non-ideal behaviour of the adsorbent as described by Peters and Bakkeren (1994). The instrument has an effective upper volume limit of 250 mL (see Section 5.1) and a lower limit of 3 mL. The volume integration becomes unreliable below 3 mL due to the additional uncertainty brought about by the dead volume before the trap (approximately 1.6 mL). On the other hand, when ambient levels

of isoprene are low (< 500 ppt), large sample volumes (200 mL) should be used. Sample volumes lower than or equal to 200 mL are used in order not to exceed the trap breakthrough volume (see Section 5.1).

The limit of detection is determined for a specific set of runs by allowing a signal-to-noise ratio (S/N) of 3. The blank runs are used to calculate the noise, which is defined as the standard deviation in the PID signal in a section of the blank chromatogram

corresponding to the isoprene elution time. The instrument response factor is calculated from the isoprene peak height in the calibration runs and the isoprene amount fraction in the standard. This allows the calculation of the minimum concentration needed to give rise to a signal that would return a S/N of 3. This is identified as the limit of detection and is monitored routinely during field deployments and laboratory tests. The limit of detection for two versions of the iDirac, the grey and the orange instruments (see Section 5.1 for details), during their deployment in Wytham Woods (See Section 5.2) were 108 ppt and 38.1

20    ppt respectively. These are higher than the limit of detection determined in the laboratory (46 ppt and 19 ppt respectively). The difference between field and laboratory sensitivity is due to greater noise in the field measurements, as a result of less controlled environmental conditions. The difference in the limit of the detection between the two instruments is attributed to differences in instrumental noise (the noise in the Orange iDirac is 10-20% greater than that from the Grey iDirac), different responses of the PIDs to isoprene, and using traps at different stages of their life cycle (refer to Section 5.3.2 and Figure 16).

### 25    5 Tests in the laboratory and field deployments

The iDirac has been tested in a series of laboratory evaluations, at a deployment at a field station in a tropical forest in Sabah, Malaysia and in a research forest in Wytham Woods, UK.

### 5.1 Laboratory Tests

**Intercomparison of two versions of the iDirac.** Two iDirac instruments (orange and grey) were compared against one

30    another, with the two instruments sampling from a chamber containing a controlled isoprene concentration which was varied over time. The orange and grey iDiracs both had inlets inside the chamber with identical filters (polyethersulfone, 0.45µm

pore-size) and the same 1.5 m length of PTFE 1/16" tubing, placed as close to one another as possible. The gas within the chamber was well mixed with two large fans. Gas from a 700 ppb isoprene (± 5%) in a nitrogen balance mixture (BOC) was flow-controlled into the chamber at 80 mL min$^{-1}$ for different time periods to change the concentration. The chamber was not flushed and the only exchange out of the chamber was slight seepage through several small holes around the inlets. The concentration was varied stepwise from 0 to 12 ppb. The instruments were calibrated using the same calibration standard (8.3 ± 0.6 ppb isoprene in nitrogen), which was connected to both instruments via a tee-piece.

The results from this experiment are shown in Figure 9. The orange iDirac under-reads by 6.6% relative to the grey iDirac, and this is particularly evident at high concentrations (> 8 ppb). Figure 10 shows this data as a scatter plot of the 15-minute average values from either instrument, again it can be seen that the orange iDirac under-reads slightly. This under-reading is partly attributed to the systematic underestimation of the peak areas from the Orange runs due to peak tailing. Integration of a subset of chromatograms using an exponentially modified Gaussian function showed that a simple Gaussian fits underestimate peak areas from the Orange instrument by up to 2 %. No significant degree of tailing was observed in the runs from the Grey instrument. Despite this slight discrepancy between the output isoprene concentration from the two instruments, it should be noted that the two iDiracs perform within their specified accuracies (see Section 4.2).

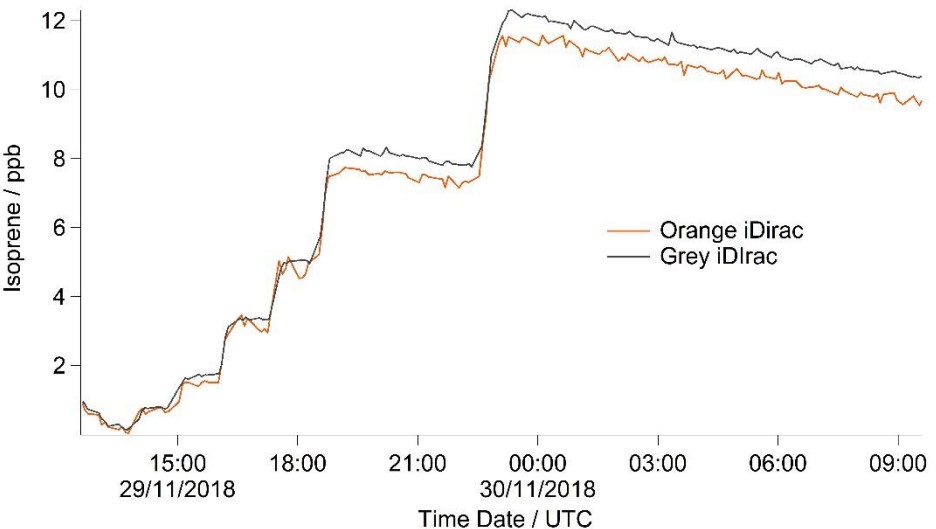

**Figure 9: Time series plot showing isoprene mixing ratios from the grey and orange iDiracs**

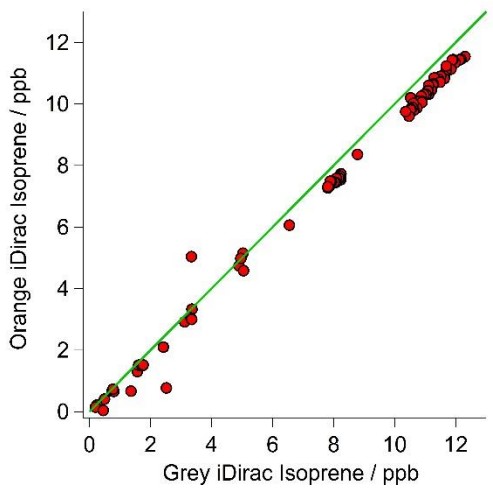

**Figure 10: Scatterplot with 1:1 line showing 15 minute average values for the grey and orange iDiracs**

**Breakthrough tests.** The breakthrough volume for the adsorbent traps used in the iDirac was determined. This is a test which evaluates what volume of gas is so great as to cause isoprene to pass through the trap in a single sample run, and is typically

independent of the analyte concentration (Peters and Bakkeren, 1994). This test is performed by placing an additional adsorbent trap in the instrument upstream of the main trap, at the exit of valves 1-4 from the valve box. Each run sampled 10 mL of a mixture of 5 ppb isoprene and 5 ppb $\alpha$-pinene in a nitrogen balance. When the breakthrough volume of the additional trap is exceeded, isoprene effectively 'breaks through' from the additional trap onto the main trap, so that it is injected onto the dual column system and a peak is observed in the chromatograms. The sum of all the volumes of the runs in which isoprene was not observed (i.e., pre-breakthrough) gives the breakthrough volume. This value effectively acts as an upper limit of the volume of gas that the instrument can sample. Figure 11 shows a typical example of such test, in which a breakthrough volume of 250 mL was determined. The instrument is therefore set to sample volumes up to 200 mL, so that the breakthrough volume is never exceeded.

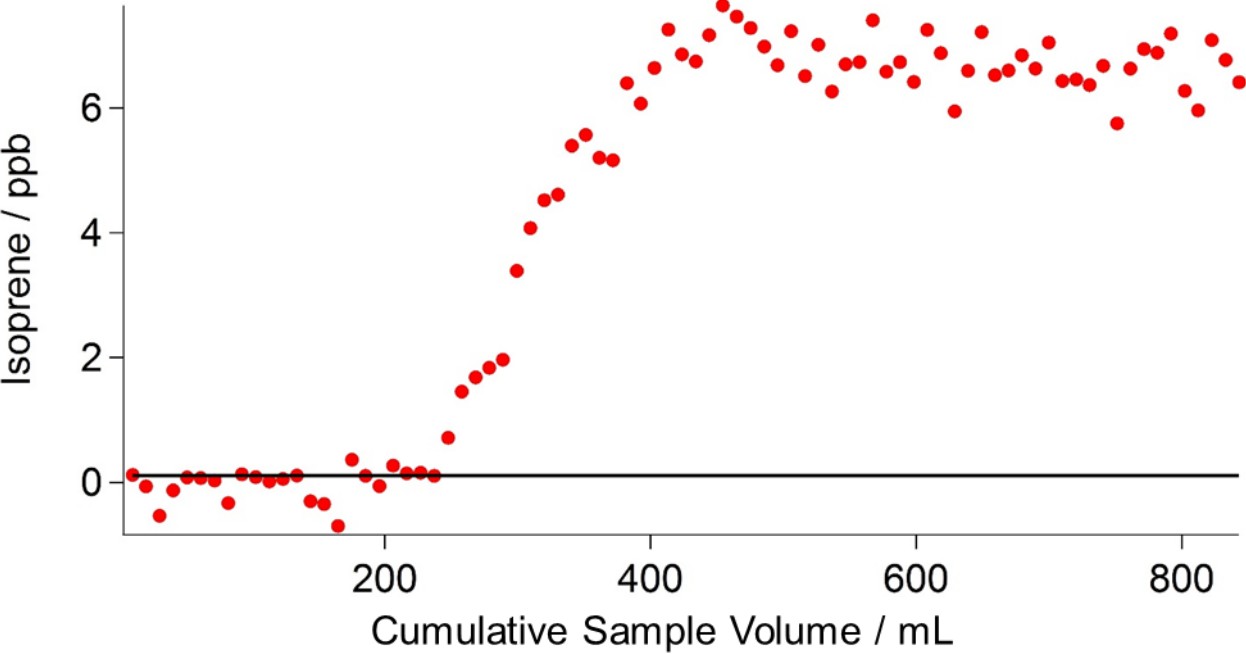

**Figure 11: Results of the breakthrough volume tests. Each data point is an individual sample run of 10 mL. A solid black line indicates a threshold (set at LOD of 0.108 ppb), above which the breakthrough volume is exceeded.**

**Co-elution of interfering species**. The PID used in the iDirac is sensitive to all molecules with ionisation energies less than or equal to 10.6 eV, which includes the vast majority of biogenic and anthropogenic VOCs with the exclusion of ethane, acetylene, propane, methanol, formaldehyde and a number of halogenated hydrocarbons. It is therefore possible that species co-eluting at the same time as isoprene might be detected and erroneously identified as isoprene, thus leading to reporting of spurious concentrations. The stationary phase in the main column is selected to achieve good separation of isoprene from VOCs of similar polarity and boiling point. This is tested in a series of co-elution experiments, in which the elution time of a number of potentially interfering species was determined and their separation from isoprene assessed. The VOCs under test were chosen based on the column specifications reported by the manufacturer, which identified i- and n-pentane, 1-pentene, trans- and cis-2-butene, 2-methyl-1-butene and 2-methyl-1-pentene as potentially co-eluting with isoprene. Gas samples containing 10-20 ppb of each interfering VOC are prepared in 3 L Tedlar bags by two-step dilutions from the "pure" substance (Sigma Aldrich, purity typically > 98%) using grade 5.0 nitrogen (purity > 99.999%). For each interfering species, the iDirac alternated between sampling from one of the Tedlar bags and sampling from a gas cylinder containing only isoprene in nitrogen. The results of these measurements are summarised in Figure 12. Figure 12a illustrates overlaid chromatograms for each species, whilst the individual chromatograms are shown in Figure 12b-h. Figure 12i summarises the different elution times taking into account the width of each peak (full-width, half maximum) to better assess separation. The isoprene peak is well separated from all interfering VOCs, while we observe poor separation between cis- and trans-2-butene (which are not

separated at all and appear as a single peak in Figure 12d) and 2-methyl-1-butene, as well as between i- and n-pentane. Similar tests were carried out for acetone and ethanol, and we found they eluted outside of the chromatographic window considered here. These results lend confidence to the unequivocal assignment of the isoprene peak in each chromatogram. Work is ongoing to determine the elution time of a wider range of compounds, including oxygenated products from the oxidation of isoprene.

Co-elution and multiple peaks appearing in a chromatogram are also addressed in the Mathematica script described in Section 3.3. To ensure that the isoprene peak is correctly assigned, the script looks for a peak in a relatively narrow region of the chromatogram, which is based on an interpolation of the elution time from the two nearest calibration runs. This algorithm has relatively low tolerance, so that peaks that are more than 4 s away from the predicted isoprene elution time are not considered. We observe a consistent discrepancy in isoprene elution time between the calibration and sample runs. The elution time of isoprene is typically 1.7 s greater in a sample run than in a calibration run. This is an artefact of the trap adsorption process and the resulting tailing of the peak. For large volumes and low concentrations (e.g., a 150 mL field sample at 0.5 ppb), the isoprene band in the adsorbent trap is very broad and resides in the trap for a longer time, so it tails very strongly. For a high-concentration low-volume sample (e.g., a 12 mL calibration run at 10 ppb), the isoprene band in the trap adsorbent is very sharp; it desorbs quickly and hence it tails less. This difference in elution times is much smaller than the distance to nearest interfering species (2-methyl-1-pentene, which elutes ~7 s before isoprene).

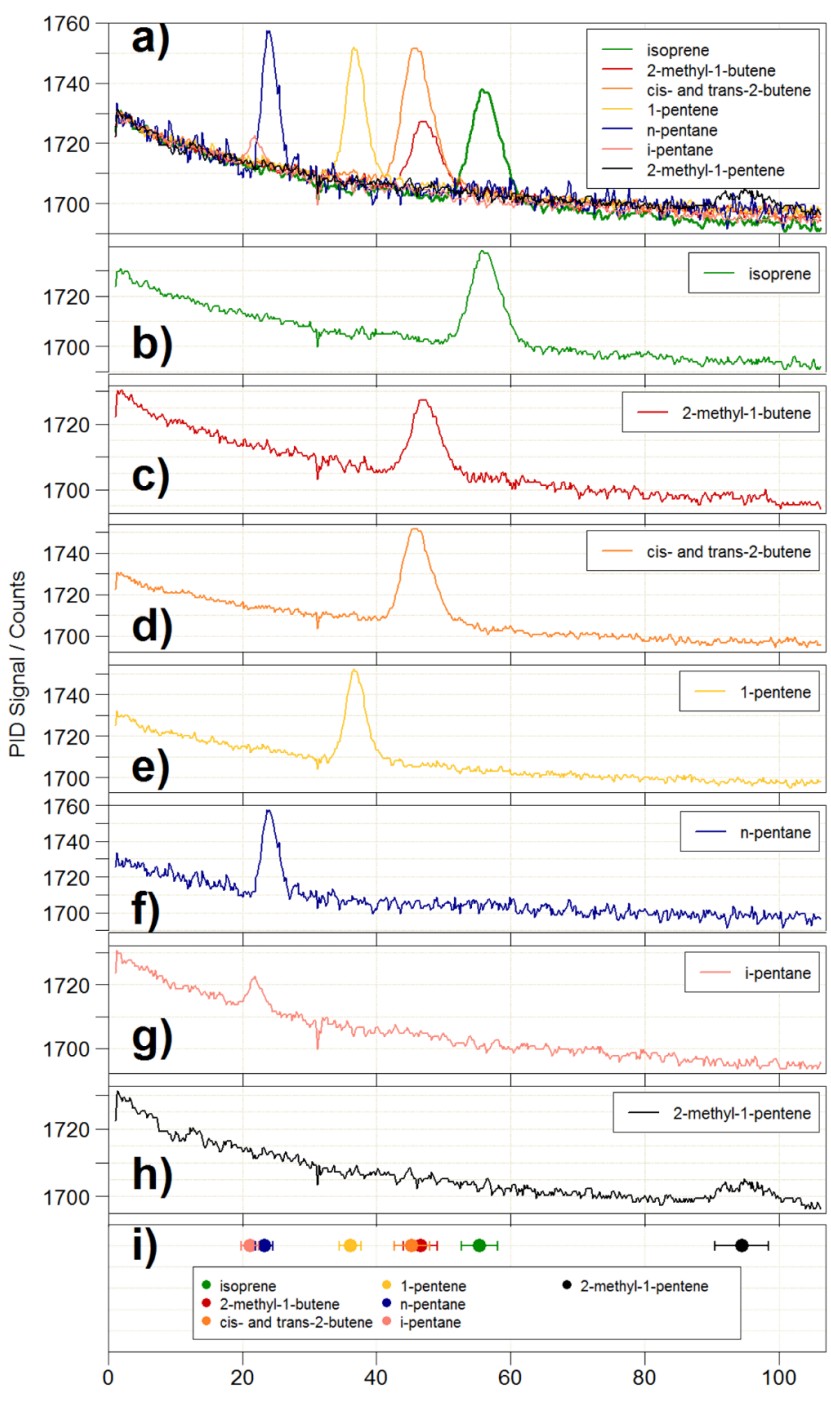

**Figure 12: Results of the co-elution tests on the iDirac. a) Overlaid chromatograms of isoprene (green line) and six potential interfering species: 2-methyl-1-butene (red line), cis- and trans-2-butene (orange line), 1-pentene (yellow line), n-pentane (blue line), i-pentane (pink line) and 2-methyl-1-pentene (black line). The chromatograms of each individual species are shown in panels b)-h). The co-elution tests are summarised in h), where the elution time of each species (filled circles) is plotted along with its peak width (FWHM, error bars) to assess peak separation.**

Peak width and RMSE from the Gaussian fit, retrieved from the fitting routine described in Section 3.3, can also be used to evaluate the presence of co-eluting species. An additional peak overlapping to some degree with the target isoprene peak in a sample run would cause a change in the peak shape. This would result in values for the fitted peak width and RMSE that are different from those from the calibration runs. For this reason, we use the width and RMSE from the calibration runs to define a range of acceptable peak widths and RMSE (equal to the mean value ± 1 standard deviation). Any peaks from sample runs exceeding this range are flagged up for further analysis.

**Long-Term Tests.** The performance of the instrument in the field for long periods of time has been assessed in several deployments. These are described in detail in Section 5.2

**5.2 Deployment of the iDirac in Sabah, 2015**

Following laboratory development and testing, the iDirac had its first field deployment at the Bukit Atur Global Atmospheric Watch (GAW) station in the Danum Valley Conservation Area in Sabah (Malaysian Borneo) as part of the Biodiversity and Land-use Impacts on Tropical Ecosystem Function (BALI) Plant Traits campaign. This campaign ran from May to December 2015. The instrument was principally used to carry out individual leaf measurements in the field. The results from the individual leaf measurements are being written up for publication elsewhere.

The other type of measurements taken in Sabah during this timeframe were longer duration runs, in which the instrument took autonomous measurements of ambient air at the field site continuously. These measurements consisted in attaching the iDirac to a tree at a height of approximately 1 m, with a battery and a 1.2 L $N_2$ cylinder attached to it, and running repeat samples until either the battery ran flat or the gas supply was exhausted. The aim of these measurements was to obtain an isoprene diurnal profile and observe how this varied with different types of forest. These tests also allowed us to test the feasibility of leaving the instrument running for long periods of time. A picture of the iDirac measuring ambient air in the rainforest in Borneo is shown in Figure 13.

The ambient air measurements demonstrated that the instrument can easily measure the changes in isoprene concentration in the ambient air and that the inlet drying system could cope with the high humidity of the rainforest. An example from a secondary forest site is shown in Figure 14. This was the first deployment for the iDirac, and it proved to be a success in taking reliable measurements. It also highlighted areas for instrument development (e.g., calibration routine) and several issues with instrument function (e.g., warm-up time) that had been addressed in subsequent versions.

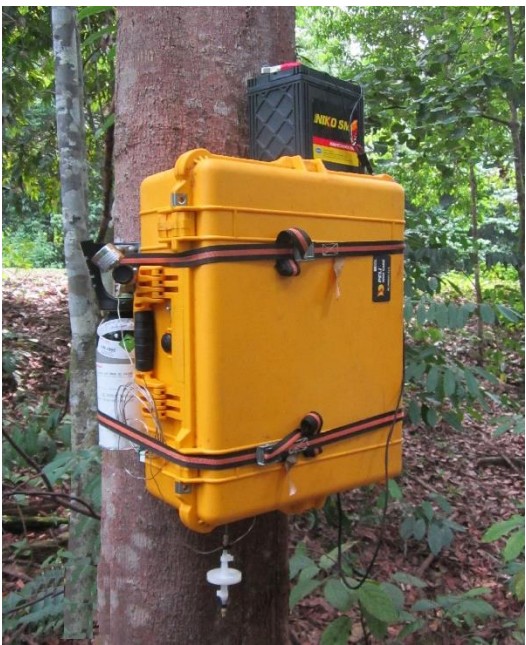

**Figure 13: iDirac deployed in a tropical forest environment**

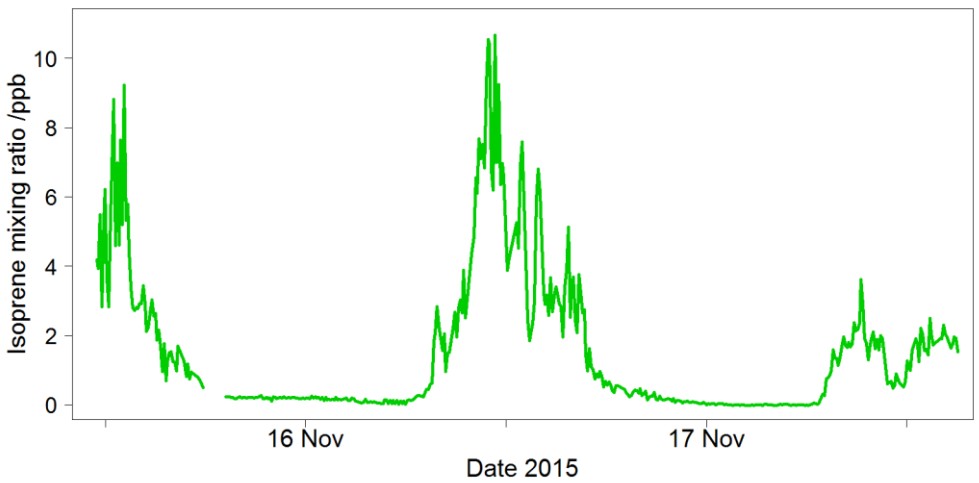

**Figure 14: Time series for isoprene at secondary forest site in Sabah (Malaysian Borneo) in 2015**

**5.3 Deployment of the iDirac in Wytham Woods (University of Oxford)**

**5.3.1 Experiment description**

The instrument was deployed at Wytham Woods (Oxfordshire, UK), a temperate mixed deciduous forest owned and managed by the University of Oxford. A large fraction of trees at this site are Pedunculate Oaks (*Quercus robur*), which are known strong isoprene emitters (Lehning et al., 1999). One iDirac was deployed on the canopy walkway facility, a platform ~15 m above ground resting on a scaffolding support allowing access to crown-level measurements, while another iDirac was installed at ground level. As each instruments has two inlets, this allowed sampling at four heights across the canopy with a view to investigate the isoprene concentration gradient within the canopy. Both iDiracs were run off-grid, powered only by solar-powered batteries. The experiment and results are described in detail by Ferracci et al. (2020) and Otu-Larbi et al. (2019). Data was collected from May – October 2018, and here the performance of the instruments is assessed for more than five months of continuous use in a forest environment.

**5.3.2 Results and discussion**

The iDirac captured isoprene concentrations from 25 May to 29 October 2018. Gaps in the data were generally due to power issues arising from insufficient solar charging of the batteries. A section of the isoprene time series is shown in Figure 15. The diurnal pattern of isoprene is clearly visible, and the vertical concentration gradient is also apparent.

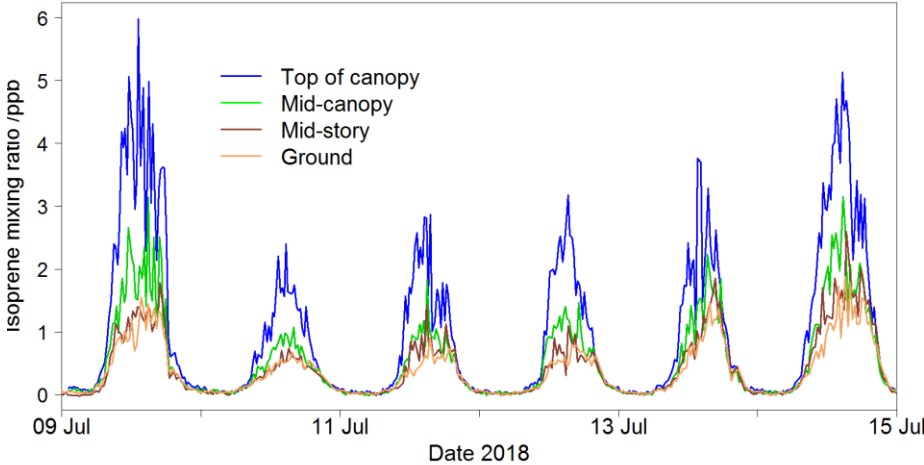

**Figure 15: Portion of the isoprene mixing ratio time series measured at Wytham Woods (UK) at four heights within a forest canopy in the summer of 2018.**

The iDirac proved capable of measuring isoprene abundances continuously through the day, spanning from concentrations as high as 8 ppb in the height of the summer and to effectively zero at night-time.

The lifetime of the absorbent trap can be assessed by examining the calibration curves over time. The dataset is analysed in weekly segments, with a calibration curve constructed for each week. This allows for the calibrated data to account for any drift in sensitivity. The calibration plots exhibit a clear drift as time progresses, as shown in Figure 16, with calibration chromatograms later in the time series showing lower peak area for the same concentration. Once the trap is replaced, higher sensitivity is recovered (shown as the green dashed line in Figure 16a and the green square in Figure 16b).

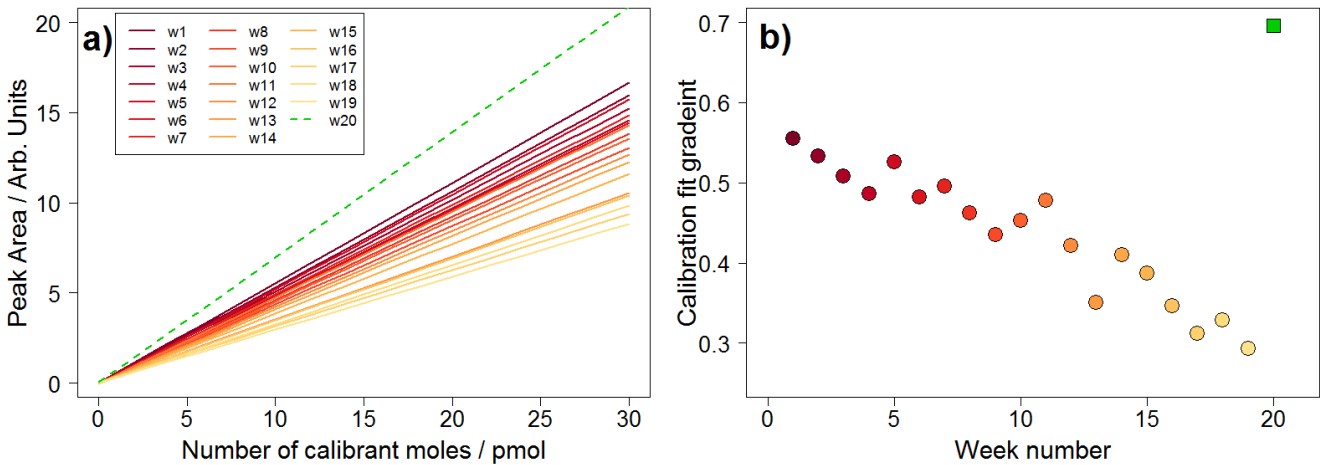

**Figure 16: a) Calibration lines plotted in weekly intervals, showing decreasing sensitivity over time (solid lines). The green dashed line is the calibration plot obtained once the trap was replaced. b) Time series of the gradients of the calibration plots showing decreasing sensitivity (filled circles). The green square represents the calibration plot gradient obtained after replacing the trap.**

This drift is attributed to the gradual degradation of the trap as a result of repeated absorption/desorption cycles, with exposure to high concentrations of VOCs and oxygen. As each week of data represents approximately 1000 absorption/desorption cycles, it is likely that the absorbent in the trap is degraded over time and eventually needs to be replaced.

Decreasing sensitivity would obviously affect the limit of detection of the instrument. During a particularly long deployment such as that in Wytham Woods, it is important to monitor the sensitivity by means of plots such as that in Figure 16 to establish better when the trap needs to be replaced.

**6 Conclusions and future work**

We described the development and subsequent deployment of the iDirac, a novel autonomous GC-PID for isoprene measurements in remote locations. The instrument pre-concentrates ambient VOCs on an adsorbent trap and then separates them in a dual column system kept in an isothermal oven before detection by a photoionisation detector, achieving a limit of detection for isoprene down to 38 ppt. The rugged design and modular construction make the instrument easily customisable, while the open source software control results in a straightforward instrument configuration. Designed for field deployments in remote environments with limited power supply, the iDirac weights 10 kg (excluding gas supply), consumes minimum power and gas, can be run autonomously for months with little maintenance (provided the performance of the trap is assessed periodically) and can be exposed to harsh environmental conditions. The sensitivity and linearity of the instrument response can be tracked effectively with regular calibrations, increasing confidence in the quality of the data. The instrument has been demonstrated to function as desired in a tropical and temperate forest in two lengthy field campaigns, in particular in summer 2018 in an Oxfordshire forest with near continuous operation for almost 6 months. While this paper focused on using the iDirac for isoprene measurements, the instrument configuration can be changed to target different analytes. Future work will focus on monitoring different VOCs (e.g., DMS and ethylene), as well as improving on some of the current limitations of the instrument, such as implementing a more sophisticated and interactive control over the oven temperature. An intercomparison in real ambient air with more established VOC monitoring instrumentation (such as that described by Barket et al. (2001)) will also help to better evaluate the accuracy of the iDirac.

**Acknowledgements**

C. Bolas acknowledges and thanks the Natural Environmental Research Council (NERC) for the Doctoral Training Partnership studentship. The research was carried out and supported with funding from the BALI project (NE/ K016377/1) . In the field we would like to thank S. Both of the University of New England, U. Kritzler of the University of Manchester, and the teams at BALI and SAFE for their support. For additional support in Borneo, many thanks to the Malaysian Meteorological Department and Universiti Malaysia Sabah. In addition, we thank Y. Malhi of the University of Oxford for help in deploying at Wytham Woods research forest. For his technical expertise and help, we gratefully acknowledge R. Freshwater at the University of Cambridge.

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
