# Peer review of "iDirac: a field-portable instrument for long-term autonomous measurements of isoprene and selected VOCs"

_Atmospheric Measurement Techniques, 2019_

## Referee Comment (RC1) · Anonymous Referee #1 · 18 Sep 2019

**Review for "iDirac: a field-portable instrument for long-term autonomous measurements of isoprene and selected VOCs"**

**General comments**

This is a well-written manuscript describing an important new instrument for quantifying atmospheric isoprene mixing ratios. In light of recent developments in remote sensing of isoprene (e.g. Fu et al. 2019), the need for a low-cost, field-deployable measurement of isoprene for satellite verification has become essential. This should be published after minor revision, as described below.

Throughout the manuscript, the authors employ a quadratic calibration curve for the instrument without sufficient discussion of the physical cause of the non-linearity, especially as they present curves with both positive and negative coefficients for the $2^{nd}$ order component of the equation (see Fig 16). In light of the overall uncertainties associated with this measurement, it would be enlightening to understand if the non-linearity is statistically significant or if a linear function would be adequate here.

What is conspicuously absent from the manuscript is a figure showing 1) a chromatogram for ambient air, to allow the reader to evaluate the validity of the peak area fitting algorithm in "real" sampling conditions and 2) an intercomparison with a well-established instrument / method for isoprene to evaluate the technique for artifact in measurement of ambient air [e.g. Barket et al., 2001]. This prevents the reader from being able to determine if the signal response of the instrument in ambient air is responding only to isoprene; because of this, the accuracy of the instrument measurement is not well-demonstrated. The underlying data presented here appears to be sufficient for the authors to provide at least two quantitative assessments of the system accuracy that are missing:

1. The diurnal profiles of isoprene shown in Figures 14 and 15 show very low observed mixing ratios for isoprene at night, indicating that species with long atmospheric lifetime relative to isoprene do not present a significant source of interference. Some statistical evaluation of the night-time / pre-dawn data may provide some bound for this potential interference to the method presented.
2. The peak-fitting technique used to determine isoprene peak area should also provide additional information, such as the fitted peak width (FWHM) and uncertainty of the peak fit area calculation. These can be used to assess the quality of the chromatographic peak, both to evaluate for co-eluters and to account for bias from changes in peak shape (e.g. tailing).

Some of the technical discussion and the number of figures seems excessive for the manuscript, and the authors should consider creating a supplemental information addendum to the main manuscript to move some of this material (see below for specific suggestions).

**Specific comments**

**P1, Line 23. Introduction.** As noted above, a recent publication [Fu et al. 2019] describes isoprene retrieval from space-based observation. The uncertainties for this method are typically 10-50%, and therefore the iDirac instrument may provide a suitable means to calibrate the satellite measurement. The above reference serves as a useful basis for defining acceptable uncertainties for an isoprene measurement, and the authors are encouraged to cite it in the introduction.

**P4, Line 12. Figure 2.** The iDirac plumbing schematic shows the sample trap as a black line between two numbers ports on the Valco valve. This resembles a valve "jumper" – a short piece of tubing connecting two ports – that is the typical convention, and can lead to some confusion. The authors should draw the sample trap as a separate device outside of the valve to make the figure clearer.

**P5, line 9. "the trap is flash-heated to approximately 300 °C"** There is no description of how this temperature is measured. Please describe the temp sensor and sensor location relative to the trap adsorbent.

**P5, line 12. "large bulky molecules"** Either "large" or "bulky" is sufficient here.

**P5, line 38. "a coiled nichrome wire heating element surrounding the section of the [stainless-steel] tube"** Is the heating element in contact with the trap tubing? It would seem that the heater would short across the trap if it is in contact. Can the authors describe how (or if) this is avoided?

**P5, line 29. "Flow restrictors upstream from valves 3, 4, 6 and 7"** The authors use small diameter tubing (0.005" ID and 0.0035" ID) rather than critical orifices to restrict reagent gas flow. Can they provide any comment on the long-term performance of this method? This reviewer uses critical orifices in this situation, which have been found to require occasional servicing.

**P5, line 39. "Carboxen 1016"** I believe the manufacturer has renamed this adsorbent to Graphsphere 2016.

**P6, line 5. "heated to 40 °C"** Can the authors provide any statistical description of the actual temperature stability of the oven? Peak retention time is directly related to this temperature, and therefore this is a critical variable in the instrument.

**P6, line 11. "so that the precolumn is back-flushed"** Is this precolumn heated above trapping temperature during the back-flush? It is not clear if the larger molecules are successfully removed during back-flush or if they accumulate.

**P6, line 30. Figure 3.** Please provide more information (i.e. isoprene mixing ratio, diluent gas, sample humidity) for this example chromatogram.

**P7, line 17. The flow through the instrument is driven by either upstream pressure (in the case of the nitrogen and calibration gas flows) or by the pump box (in the case of Samples 1 and 2).** Is the sample pressure measured? From the description here and in Figure 2 (and associated text) it is not clear if the adsorbent trap experiences higher pressure when the cal / blank solenoid valves are actuated, versus when using sample inlets 1 / 2. It is the reviewer's experience that adsorbent trapping efficiency has a pressure-dependence. Has this been observed by comparison of calibrant gas addition via the cal port versus via a sample port?

**P7, line 40. "controls the altimeter pressure sensor"** Is this the same as the differential pressure sensor described previously (P7, line 21)? And does the Arduino board control this sensor, or just read it?

**P8, line 22. "Figure 5 shows a flow diagram"** This figure could be moved to a supplemental materials section to conserve space in the main document.

**P9, line 2. "next step is to locate the isoprene peak and to fit a Gaussian curve"** It is not clear how this is performed, and could use some more explicit description. Is the user manually locating the peak or is the software algorithm doing this? How is the baseline for the Gaussian curve defined? Is there any evaluation of the goodness of the Gaussian fit to the data? There are instances in data figures throughout the text where negative peak areas or mixing ratios are shown. Is this from a fitted Gaussian with negative peak height?

**P9, line 6. "A quadratic curve is fit to this data, which captures any slight deviations from linearity."** As noted above, the use of a non-linear fit to instrument response should have a physical explanation, especially as both positive and negative coefficients are shown in the text. What is the uncertainty of the second-order coefficient? What amount of additional uncertainty would be added the overall measurement by simply using a linear fit of the calibration data?

**P9, line 12. "The Gaussian function has certain boundaries set, to further ensure that it is fitted to the correct peak."** Please describe exactly which boundaries are set. How is the Gaussian fitting function constrained?

**P9, line 15. "it can be estimated using the column temperatures"** This implies that column temperature is not constant. The temperature variance in ambient sampling should be described (see comment above).

**P10, line 1. "This type of treated cylinder exhibits very good long-term stability for a number of VOCs (Gary Barone et al. Restek Corporation, 2010)."** I don't understand this reference, and it is not listed in the Reference section.

**P10, line 2. "The exact isoprene amount fraction in the secondary standard is determined by validating it against the NPL primary standard."** Can the authors make a statement about the stability of their secondary standards over time? This

reviewer has found that secondary isoprene standards made in Aculife [Air Liquide] cylinders can degrade over the timespan of months.

**P10, line 5. "mixing rations"** should be mixing ratios

**P10, lines 10-20. "Calibration frequency is specified by the user . . ."** While the explicit description of the instrument calibration method is welcome, it is probably more appropriate to move this material, along with Figure 6, to a supplemental materials section.

**P10, line 26. "a random mixture of 3, 6, 12, 24 and 48 mL samples"** It appears that the calibration sample volumes used are always significantly smaller than the ambient air sample volumes. Since breakthrough volume is a critical parameter of the sample trap (Section 5.1 of text), I am curious as to why the calibration does not include a sample volume larger than ambient volume.

**P10, line 30. "The equation for the quadratic fit allows the determination of the fractional isoprene amount in the samples by extrapolation or interpolation"** The use of a quadratic calibration curve with extrapolation seems especially susceptible to the introduction of additional error in reporting mixing ratios.

**P11, line 1. Figure 6.** The linear fits of the peak areas for the individual calibration volumes versus time do not seem appropriate for this figure, since they are not used in the calibration procedure described in the text (peak area versus calibration sample volume on a weekly time scale) nor discussed in the text. The peak areas of 48mL calibration samples show a significant decay with time that is not apparent in the smaller calibration volumes. Is this statistically significant, and, if so, does this imply breakthrough for large sample volumes?

**P11, line 9. "The x-axis ('Effective Calibration Concentration') consists in the calibration volume (in mL) multiplied by the isoprene concentration in the gas standard (in ppb)."** Isn't this simply calibrant mass or moles, then? Are you attempting to convert these calibration points to equivalent mixing ratio in ambient air? After solving for the 'Effective Calibration Concentration' based upon ambient peak area, does the user then divide this 'Effective Calibration Concentration' by the sample volume to determine the ambient mixing ratio? The authors should provide the equation for converting peak area and sample volume to mixing ratio.

**P11, line 15. "The precision of the instrument was determined as the relative standard deviation in isoprene peak area from calibration chromatograms with the same user-specified volume . . ."** In light of the temporal trend in sensitivity discussed later in the text (5.3.2), does this overestimate the precision uncertainty? The authors do not define the length of time that is averaged here, but it appears that the instrument sensitivity decreases on the time-scale of weeks, and therefore we would not expect the peak area here to be a constant.

**P12, line 4. "The accuracy of the instrument is dictated primarily by the uncertainty in the isoprene amount fraction in the NPL standard, and how this is propagated to the isoprene amount fraction in the secondary gas standard used in the field."** Because the authors have not fully demonstrated that this technique is adequately selective to isoprene, I don't think this statement is correct. Certainty the uncertainty of the calibrant mixing ratio is a component of accuracy, but perhaps not the primary driver. The intercomparison of two iDirac instruments presented in 5.1 would indicate this as well.

**P12, line 9. "XLGENLINE, a generalised least-squares (GLS) software package for low-degree polynomial fitting (Smith, 2010) is used to estimate the final uncertainty in the isoprene amount fraction in the secondary calibration cylinder by inverse regression from the calibration curve. For most secondary calibration cylinders, this is estimated to ~ 7% at the k = 2 level (providing a coverage probability of approximately 95%). A similar procedure is applied to calibration and sample data from the field to estimate the uncertainty in the ambient isoprene concentrations. This is estimated to ~20-25 % at the k = 2 level."** I found this discussion to be difficult to assess, and reads as simply reporting the output of a statistical software package. Can the authors put this into more explicit terms, e.g. describe what is meant by "inverse regression from the calibration curve"? Also, the comment "most calibration cylinders" is vague – how often do secondary cylinders fall outside this accuracy range? It's not clear how this technique is extended to the field data; again, please describe the analysis in detail. Finally, it should be noted that the authors here use a 2-σ

uncertainty to define accuracy, but a 1-σ uncertainty to define precision above. One uncertainty level should be used for both.

**P12, line 15. Figure 8.** The test cylinder mixing ratio has a negative mixing ratio at the start of the experiment shown. Is this real data, or just an artifact of the analysis? Perhaps the data plot should start after the first standard peak area, at 21:00, 21-Feb-2018.

**P12, line 19. ". . . the high concentration of isoprene would risk *poisoning* the adsorption trap."** Italic emphasis is mine. Did you intend to use the work "poisoning" here? I interpret that this to mean an irreversible change to the adsorptive strength of the trap, while I believe that you mean that the trap would demonstrate breakthrough due to non-ideal behavior, as described by Peters and Bakkeren (1994).

**P12, line 31. "This is identified as the limit of detection and is calculated for two versions of the iDirac, the grey and the grange instruments."** I found this discussion a bit hard to follow since results from these two instruments were presented without introduction. A sentence or two from the later discussion of these instruments in Section 5.1 would be helpful here. Alternatively, simply cite the LOD for the grey instrument here and discuss the higher LOD observed for the orange instrument in section 5.1.

**P12, line 32. "This difference is attributed to the traps used (i.e., a trap with more adsorbent would retain more analyte, resulting in a larger signal)"** Doesn't this imply that the traps are not quantitative? If the amount of analyte retained is proportional to the adsorbent mass, this means the traps have a consistent breakthrough, rather than a critical sample volume as described by Peters and Bakkeran (1994). Data provided in the next section (e.g. Figure 11) shows this not to be the case. This explanation does not seem reasonable.

**P13, line 18. "This under-reading is likely due to differences in the absorbent trap, leading to a lower sensitivity for the orange instrument. This is supported by the calibration curve for the orange iDirac, which curves more at high concentrations, resulting in lower peak height than in the grey iDirac for the same concentration. Another artefact of this is that the noise visible on the orange output is greater."** I am not persuaded by this argument. Why wouldn't calibration account for these differences, irrespective of the curvature of the calibration? Applying a straight-edge to the scatterplot of the two instrument responses shown in Figure 10, I see no significant curvature, i.e. the 6.6% difference appears consistent across the mixing ratio range. See also previous comment for my concern that the adsorbent trap mass is used to explain accuracy differences. The authors may wish to consider alternative explanations, e.g. if the pressures during calibration are the same in both instruments (see comment for P7, line 17).

There are two qualitative statements presented here that should be quantified: "curves more at high concentrations" and "noise visible on the orange output is greater."

Finally, it should be noted that the two instruments perform within the specified accuracy for this system (7%), and therefore the performance demonstrated here is acceptable.

**P14, line 4. Breakthrough tests.** The results presented in this section present a very nice demonstration of the adsorbent trap showing ideal behavior with a (presumed) single-component isoprene mixture. The citation for this experiment [Peters and Bakkeren, 1994] observed that the breakthrough volume observed in this sort of experiment will be higher than when a using a test mixture that includes other compounds, which "may have a pronounced influence on the BTV and thereby on the safe sampling volume of single compounds." The authors should specify if the test mixture included other species expected in ambient air. If it did not, they should consider how valid the results presented here will be for collection of ambient air.

**P14, line 7. "Each run sampled 10 mL"** Did you mean this? Since you specify varying sample volume in figure 11, I assume this to be a typo and should read "sampled 10 mL min$^{-1}$."

**P14, line 8. "isoprene mixture of known concentration"** Perhaps better to say "isoprene mixture at constant concentration."

**P14, line 16. Co-elution of interfering species.** I found this section to be a bit trivial, due to the somewhat arbitrary list of species used for this test. I wonder if many of the alkenes used for this work (e.g. 2-methyl-1-pentene) would be

expected to be found in ambient air at significant mixing ratios to be of importance as an interferent. The authors note "Work is ongoing to determine the elution time of a wider range of compounds, including oxygenated products from the oxidation of isoprene." Since the PID is sensitive to many species beyond alkenes (e.g. ketones, aldehydes), I'm not sure if the work here is very conclusive in demonstrating that there would be no interference when measuring ambient air. I would suggest moving this section to a supplemental materials section, or revising with a more exhaustive list of species.

**P15, line 23. "This is an artefact of the trap adsorption process and the resulting tailing of the peak."** This is a nice explanation of the observed phenomenon. Have the authors considered if the tailing for peaks in ambient air relative to peaks from calibration runs leads to a bias in the peak areas? That is, a Gaussian fit function may miss some of the peak tail for ambient runs relative to calibrations, leading to consistent under-reporting of ambient mixing ratio. This effect could be tested by integrating a subset of peaks to capture all peak area, or by using an exponentially-modified Gaussian fit function.

**P16, line 11. "Sabah (Malaysian Borneo)"** Could the authors be a bit more specific for this location? Sabah is roughly the same size as Scotland.

**P17, line 7. "Error! Reference source not found.."** I believe this is a reference to Figure 13.

**P17, line 15. Figure 14.** For the time series shown, there is a significant time period (16-Nov-2015, 19:00 - 2200) where the mixing ratio trace is significantly below zero. Is this correct or a plotting error? Is this attributable to the "several issues with instrument function"? If the latter, perhaps this time period should be referenced in the text.

**P18, line 12. "5.23.2 Results and discussion"** Should read 5.3.2 Results and discussion.

**P18, line 17. Figure 15.** There is a drop in the mid-canopy time series on the morning of 7-Nov-2018 that seems anomalous with the other mixing ratio traces and other days shown. The authors may want to confirm that those data pass quality assurance checks.

**P19, line 2. Figure 16.** The calibration curves presented in this figure give me particular concern, as there appear to be consecutive weeks where there is a positive and then negative curvature. Does this seem physically possible? Generally, I found this figure difficult to interpret as I am color-blind and the color scale to distinguish weeks is quite subtle to me. If the underlying data were re-fit with a linear calibration function and a time-series of the slope were presented instead, I suspect that this figure would be much more effective. The x-axis "Effective Calibration Concentration" should have units of mass or moles, as noted for Figure 7.

**P19, line 3. "This drift is attributed to the gradual degradation of the trap"** How do the authors separate loss of trapping efficiency from loss of detector sensitivity? If a new sample trap returned sensitivity to the Week 1 level, that would be a useful data point to show.

**P19, line 26. "a more sophisticated and interactive control over the oven temperature."** This statement begs for a presentation of underlying data (oven temperature time trace, retention time drift) to justify this need. Possibly, either could be added to field data time traces in Figure 6, 14 and/or 15 on a secondary axis.

**P20, line 9. References.** At least two references (Allen et al., 2018; Helmig et al., 1998) are incomplete. The Barone et al., 2010 reference from the text is missing; see previous comment for P10, line 1.

**References**

Barket et al., J. Geophys. Res., 106: D20, 24301-24313 (2001), doi.org/10.1029/2000JD900562
Fu, D. et al., Nature Communications, 10: 3811 (2019), doi.org/10.1038/s41467-019-11835-0
Peters and Bakkeren, Analyst, 119, 71-74 (1994), doi.org/10.1039/AN9941900071

---

## Referee Comment (RC2) · Anonymous Referee #2 · 22 Oct 2019

"iDirac: a field-portable instrument for long-term autonomous measurements of isoprene and selected VOCs"

This paper thoroughly details the trial of a deployable instrument for remotely measuring isoprene (the main focus of the manuscript) and similar VOCs. There is a great need for calibrated portable instruments that can monitor long term trends out in the field for campaigns and background measurements therefore the iDirac instrument is of great interest, as this can complement reference materials and satellite measurements to give a holistic overview of the atmospheric isoprene distribution.

The paper thoroughly introduces the instrumentation and shows the breadth of mea-

surements that have been performed to test it. However, despite the wealth of information specific quantification is often not provided (please see Specific comments). I recommend that this manuscript be accepted once these minor corrections have been addressed.

General comments In the introduction it would be useful to explain the importance of isoprene with regards to impact on the OH reservoir in the troposphere, as well as on SOA. I think that a bit more detail is required about the instrumentation, particularly with regards to the drier and breakthrough and poisoning of the trap. Have any tests been performed to asses if the drier removes any VOCs? How do you recondition the trap, what if terpenes or heavy components stick and reduce trapping ability? Figure 9 – breakthrough effected by flow rate – how were the samples pumped into the system – what happens if you vary the sampling rate? Similar questions arise from the results depicted in Figure 11. It would also be helpful to have a clearer idea of what is meant by the trap becoming "poisoned" is this the result of moisture and what is the impact on uncertainties and sensitivity? I strongly suggest that Figure 3 includes information about the isoprene concentration depicted and I would also like error bars added to all figures where required e.g. Figure 7. Volumes should have an associated uncertainty. Finally, I think it would be useful to know why was a Gaussian shape was used for the fitting, have you tried any other peak shapes for fitting e.g. Voigt or speed-dependent Voigt? These might optimise results.

Specific comments P1. Line 23. Please change to "Isoprene is an important non-methane"... P1. Line 25. What is the impact of the SOA, please also add references P2. Line 16. Grab samplers: please add example reference e.g. Robinson AD DOI: 10.5194/acp-5-1423-2005 P2. Line 36. What about trueness? P3. Table 1. Please specify nitrogen purity percentage P4. Figure 1. Does the packaging and foam emit any VOCs? P5. Paragraph 2. What is the volume sampled? P5. Line 30. Why can you not use a non-return valve? P5. Line 43. Specify the desired volume P6. Lines 11-14. Consider combining sentences. P6. Line 21. Are there any other VOCs with

similar ionisation values or boiling points? P6. Figure 3. What concentration does this peak represent? What's the S/N ratio? P7. Line 7. Should read "in a nitrogen balance" P7. Line 9. Is the calibration gas purchased or decanted, if the latter please specify how. P7. Line 20. Please specify the reduced pressure P7. Line 23. I think that you mean "nominally" not "typically"? P7. Line 38. IS the clock calibrated? P9. Line 15. What are the criteria for insufficient/sufficient (how many?) P9. Line 16. What is the criteria for "too great" P9. Line 26. What is the stability period of the gas standard? P10. Line 13. Replace "good practice" with "essential"! P10. Line 14-19. Please give numbers. P10. Line 26. What is the error on the volumes? P11. Paragraph 1. Are we talking about intermediate precision or reproducibility? P12. Line 13. What is the lowest volume used – this may impact uncertainty and S/N as the sensitivity is likely to vary with volume size P12. Line 31. "the grey and the grange instruments" I think you mean "orange"; please clarify (explain what the difference is before section 5.5). P13. Line 11. Specify uncertainty of BOC mixture and add the word "balance" to describe the matrix gas. P13. Line 12. Do you have an offset from losses to the chamber wall? P13. Line 15. Should be "tee-piece" P13. Line 17. "high" Specify above 8 ppb. P13. Line 22. What happens if you switched the trap? Have you considered the impact of breakthrough at high flow rates? P15. Line 8. Rephrase "pure substance". Surely there will be impurities in the raw materials? Perhaps state the purity of the reagent? P16. Figure 12. What does a blank run look like? P17. Line 7. Please resolve "Error! Reference source not found..." P17. Line 12. How has this been addressed in subsequent versions? P19. Figure 16. Can you please use a wider colour range? P19. Line 4. Is the poisoning moisture? What impact does this have on the sensitivity and uncertainties attributed? P19. Line 20. "can be run autonomously for months" Assuming the trap is not degraded?

---

## Author Comment (AC1) · 3 Jan 2020

Reviewers' comments are in black, authors' response in blue and changes to the text in red.

**Reviewer #1**
**General comments**

This is a well-written manuscript describing an important new instrument for quantifying atmospheric isoprene mixing ratios. In light of recent developments in remote sensing of isoprene (e.g. Fu et al. 2019), the need for a low-cost, field deployable measurement of isoprene for satellite verification has become essential. This should be published after minor revision, as described below.

We thank the reviewer for their valuable feedback on the manuscript. Below we address the individual points raised by the reviewer.

Throughout the manuscript, the authors employ a quadratic calibration curve for the instrument without sufficient discussion of the physical cause of the non-linearity, especially as they present curves with both positive and negative coefficients for the 2nd order component of the equation (see Fig 16). In light of the overall uncertainties associated with this measurement, it would be enlightening to understand if the non-linearity is statistically significant or if a linear function would be adequate here.

We investigated the impact of using linear or quadratic calibration curves on the isoprene mixing ratios in the sample runs, and found only minor differences (< 1%). This is because the $2^{nd}$ degree coefficient (i.e., the curvature) is effectively very small, and all the sample measurements presented here fall within the dynamic range of the calibration plots (i.e., all determined by interpolation and not extrapolation). The calibration lines in Figure 16 gave an exaggerated idea of the curvature as they effectively extrapolated beyond the highest calibration point (i.e., outside the dynamic range of both calibration and sample runs). We have also amended the x-axis in Figure 16 to reflect this.

Upon closer inspection we found that the statistical significance of the curvature is small even without taking into account the uncertainty in the measurements (i.e., just from the scatter of the points in the calibration plots), and is indeed negligible once the measurement uncertainty (~10%) is taken into account. We also found that, by varying the length of the time interval over which calibration runs are used to produce a calibration plot (typically 1 week; in this test varied between 4 to 10 days), the sign of the curvature of the calibration plot that would be applied to a few selected sample chromatograms changed sign. We conclude that the curvature itself is not an inherent feature of the instrument over the calibrant mass used and that the sign of the curvature is predominantly determined by the number of calibration points at high volumes, as this varies from calibration plot to calibration plot. We agree that a linear fit is better suited to these plots and have amended the plots and text accordingly.

What is conspicuously absent from the manuscript is a figure showing 1) a chromatogram for ambient air, to allow the reader to evaluate the validity of the peak area fitting algorithm in "real" sampling conditions and 2) an intercomparison with a well-established instrument / method for isoprene to evaluate the technique for artefact in measurement of ambient air [e.g. Barket et al., 2001]. This prevents the reader from being able to determine if the signal

response of the instrument in ambient air is responding only to isoprene; because of this, the accuracy of the instrument measurement is not well demonstrated.

In response to point 1), we now present a calibration, a sample and a blank chromatogram with peak fit in the revised Figure 3.

We agree with the reviewer regarding point 2). We are attempting to arrange an intercomparison with a more established instrument for isoprene monitoring under ambient conditions. We have added a sentence to address this point in the Conclusions section to reflect this:

An intercomparison in real ambient air with more established VOC monitoring instrumentation (such as that described by Barket et al. (2001)) will also help to better evaluate the accuracy of the iDirac.

The underlying data presented here appears to be sufficient for the authors to provide at least two quantitative assessments of the system accuracy that are missing:

1.  The diurnal profiles of isoprene shown in Figures 14 and 15 show very low observed mixing ratios for isoprene at night, indicating that species with long atmospheric lifetime relative to isoprene do not present a significant source of interference. Some statistical evaluation of the night-time / pre-dawn data may provide some bound for this potential interference to the method presented.

Assuming there are long-lived co-eluters that are trapped with the same efficiency and have the same PID response as isoprene, the night-time observed concentrations attributed to isoprene could be thought of as an upper limit to this type of interference. We calculated the mean night-time isoprene mixing ratio from the plots in Figures 14 and 15 as 50 ppt in both cases. We have added the following paragraph at the end of the subsection on accuracy in Section 4.2:

"Co-elution of interfering species can also affect accuracy. Tests targeting specific potential interferents are described in Section 5.1 and show that these species do not overlap with the isoprene peak in the chromatograms. However co-elution with unknown (or not tested for) species, albeit unlikely, can never be fully ruled out. If these species have longer lifetimes than isoprene, the observed night-time abundances attributed to isoprene can be used as the upper limit of potential interference of unknown co-eluters (assuming they are trapped with the same efficiency and have the same PID response as isoprene). The isoprene night-time mixing ratio is 50 ppt for the data shown in both Figures 14 and 15. Therefore we estimate the instrument accuracy for field data as the combination of the propagated uncertainty from the standard (10-12.5 %) and the potential co-elution of long-lived species (50 ppt). This correspond to an overall accuracy of ± 1.2 ppb for a 10 ppb isoprene sample, ± 0.13 ppb for a 1 ppb isoprene sample and ± 51 ppt for a 100 ppt isoprene sample."

2. The peak-fitting technique used to determine isoprene peak area should also provide additional information, such as the fitted peak width (FWHM) and uncertainty of the peak fit area calculation. These can be used to assess the quality of the chromatographic peak, both to evaluate for co-eluters and to account for bias from changes in peak shape (e.g. tailing).

We monitor peak features throughout our analysis routine and have added the following sentence to the end of the co-elution section to address this point:

"Peak width and RMSE from the Gaussian fit, retrieved from the fitting routine described in Section 3.3, can also be used to evaluate the presence of co-eluting species. An additional

peak overlapping to some degree with the target isoprene peak in a sample run would cause a change in the peak shape. This would result in values for the fitted peak width and RMSE that are different from those from the calibration runs. For this reason, we use the width and RMSE from the calibration runs to define a range of acceptable peak widths and RMSE (equal to the mean value ± 1 standard deviation). Any peaks from sample runs exceeding this range are flagged up for further analysis."

Some of the technical discussion and the number of figures seems excessive for the manuscript, and the authors should consider creating a supplemental information addendum to the main manuscript to move some of this material (see below for specific suggestions).

**Specific comments**

**P1, Line 23. Introduction**. As noted above, a recent publication [Fu et al. 2019] describes isoprene retrieval from space based observation. The uncertainties for this method are typically 10-50%, and therefore the iDirac instrument may provide a suitable means to calibrate the satellite measurement. The above reference serves as a useful basis for defining acceptable uncertainties for an isoprene measurement, and the authors are encouraged to cite it in the introduction.
We added a paragraph to the introduction, reading as follows:
Recent work showed that it is possible to retrieve isoprene abundances in the boundary layer using satellite measurements by means of thermal infra-red imaging (Fu et al., 2019). However, with uncertainties in the range of 10-50%, these retrievals would benefit from further validation from ground-based instrumentation.

**P4, Line 12. Figure 2**. The iDirac plumbing schematic shows the sample trap as a black line between two numbers ports on the Valco valve. This resembles a valve "jumper" – a short piece of tubing connecting two ports – that is the typical convention, and can lead to some confusion. The authors should draw the sample trap as a separate device outside of the valve to make the figure clearer.
Amended.

**P5, line 9. "the trap is flash-heated to approximately 300 °C"** There is no description of how this temperature is measured. Please describe the temp sensor and sensor location relative to the trap adsorbent.
The trap temperature is measured routinely in the lab but not in the field. We found the temperature varied by ± 5°C. We have added this to the text.

**P5, line 12. "large bulky molecules"** Either "large" or "bulky" is sufficient here.
Amended.

**P5, line 38. "a coiled nichrome wire heating element surrounding the section of the [stainless-steel] tube"** Is the heating element in contact with the trap tubing? It would seem that the heater would short across the trap if it is in contact. Can the authors describe how (or if) this is avoided?
The nichrome wire has a ceramic electrically insulating coating to prevent shorting with the trap tubing. We have added a sentence to the text to reflect this.

**P5, line 29. "Flow restrictors upstream from valves 3, 4, 6 and 7"** The authors use small diameter tubing (0.005" ID and 0.0035" ID) rather than critical orifices to restrict reagent gas flow. Can they provide any comment on the long-term performance of this method? This reviewer uses critical orifices in this situation, which have been found to require occasional servicing.

We found good performance in flow regulation over time. Naturally the flow rate downstream of the restrictors is proportional to the gas pressure upstream of the restrictors. However, if the upstream pressure is not changed, we found very little variation in the flow rate. For instance, the plot below shows the flow rate of the calibration gas (valve 3) across a 2-month period in the WIsDOM campaign (described in Section 5.3) during which the regulated output pressure of the calibration cylinder was not changed. The flow (green points) remained stable, within 2.5% (dashed red lines, corresponding to 1 standard deviation) from the average (solid red line).

[Figure]

**P5, line 39. "Carboxen 1016"** I believe the manufacturer has renamed this adsorbent to Graphsphere 2016.

Amended.

**P6, line 5. "heated to 40 °C"** Can the authors provide any statistical description of the actual temperature stability of the oven? Peak retention time is directly related to this temperature, and therefore this is a critical variable in the instrument.

The oven temperature exhibits diurnal variations (typically, ± 2 °C) that appear driven by ambient temperature. This in turn affects the isoprene retention time. However, with frequent calibrations (every 5 hours) it is possible to track how the retention time changes with time. We found that interpolating between the elution times from two adjacent calibration runs was sufficient to determine the elution time in the sample runs in between. When calibration runs are more infrequent, we derive a linear relationship between oven temperature and isoprene retention time from the calibration runs to constrain the range over which the algorithm fits a Gaussian in the sample runs.

We have added the following sentence to Section 2.1:

"The oven temperature exhibits diurnal variations (typically in the range of ±2 °C) that appear driven by ambient temperature. This introduces some variability in the isoprene retention time, but it is accounted for in the analysis of chromatograms (see Section 3.3)."

**P6, line 11. "so that the pre-column is back-flushed"** Is this pre-column heated above trapping temperature during the backflush? It is not clear if the larger molecules are successfully removed during back-flush or if they accumulate.

The pre-column is inside the oven along with the main column, and the oven (hence both columns) is always kept at a higher temperature than the rest of the Pelicase, where the trap is located. As we log the temperature inside the Pelicase at the end of the trapping step, we can look at the difference between oven temperature and trapping temperature. The oven temperature is always 4-8 °C higher than the temperature in the rest of the Pelicase.
We do not see any signs of carryover between runs, and especially in the blank and calibration runs, so we conclude that larger molecules are successfully removed by the backflush.

**P6, line 30. Figure 3**. Please provide more information (i.e. isoprene mixing ratio, diluent gas, sample humidity) for this example chromatogram.

Figure 3 has been amended include a calibration, a sample and a blank chromatogram with the corresponding fits to the data. The caption has also been amended to include the information requested.

**P7, line 17. The flow through the instrument is driven by either upstream pressure (in the case of the nitrogen and calibration gas flows) or by the pump box (in the case of Samples 1 and 2)**. Is the sample pressure measured? From the description here and in Figure 2 (and associated text) it is not clear if the adsorbent trap experiences higher pressure when the cal / blank solenoid valves are actuated, versus when using sample inlets 1 / 2. It is the reviewer's experience that adsorbent trapping efficiency has a pressure-dependence. Has this been observed by comparison of calibrant gas addition via the cal port versus via a sample port?

Whilst sample pressure is not measured during the instrument routine operation, we have measured the gas pressure upstream of the trap (and downstream of the solenoid valves), and found no difference in pressure between sample, calibration or blank runs. This indicates that the flow restrictors also contribute to regulating the pressure down to ambient. We carried out further experiments in which the same mixture was fed into the instrument at different pressures (1.5-2.5 bar gauge) through valve 3 (calibration port), and also at 0.1 bar gauge through valve 1 (sample 1 port) and found no difference in the resulting integrated peak area, as shown below (error bars indicating 1 standard deviation of at least 10 replicates at each pressure).

[Figure]

**P7, line 40. "controls the altimeter pressure sensor"** Is this the same as the differential pressure sensor described previously (P7, line 21)? And does the Arduino board control this sensor, or just read it?

No, this is a different pressure sensor, and effectively measures ambient pressure. The Arduino board only reads it, and the text has been changed accordingly.

**P8, line 22. "Figure 5 shows a flow diagram"** This figure could be moved to a supplemental materials section to conserve space in the main document.

We feel this figure helps understanding the processing of the chromatogram and wish to retain it in the main text.

**P9, line 2. "next step is to locate the isoprene peak and to fit a Gaussian curve"** It is not clear how this is performed, and could use some more explicit description. Is the user manually locating the peak or is the software algorithm doing this? How is the baseline for the Gaussian curve defined? Is there any evaluation of the goodness of the Gaussian fit to the data? There are instances in data figures throughout the text where negative peak areas or mixing ratios are shown. Is this from a fitted Gaussian with negative peak height?

We have replaced that sentence with the following text to clarify how the peak is located and fitted to:

"From the plot of all calibration chromatograms, the user specifies the regions that are used to fit to the isoprene peak and the baseline. A third-degree polynomial is fitted to the baseline over the user-specified baseline intervals. A Gaussian curve is then fitted to the baseline-subtracted chromatogram over the user-specified peak interval."

Regarding the evaluation of the goodness of the fit, we retain the RMSE from the fit for each chromatogram (refer to general comment 2 on page 2 for this is used to evaluate sample peaks). The height of the Gaussian is allowed negative values, but this only occurs when isoprene is near or below the detection limit (i.e., blank or night-time runs), or during instrument warm-up. The instances of negative areas (and mixing ratios) mentioned by the reviewer have been reanalysed and amended.

**P9, line 6. "A quadratic curve is fit to this data, which captures any slight deviations from linearity."** As noted above, the use of a non-linear fit to instrument response should have a physical explanation, especially as both positive and negative coefficients are shown in the text. What is the uncertainty of the second-order coefficient? What amount of additional uncertainty would be added the overall measurement by simply using a linear fit of the calibration data?

This has been amended to a straight line fit. Refer to the answer to the first comment on page 1 for further details.

**P9, line 12. "The Gaussian function has certain boundaries set, to further ensure that it is fitted to the correct peak."** Please describe exactly which boundaries are set. How is the Gaussian fitting function constrained?

The text now reads:

"Then a Gaussian function, constrained by certain boundaries (e.g., peak width within the average calibration peak width ± 1 standard deviation, retention time within ±5% of the interpolated retention time), is fitted to the section of the chromatogram indicated by the interpolated calibration retention times."

**P9, line 15. "it can be estimated using the column temperatures"** This implies that column temperature is not constant. The temperature variance in ambient sampling should be described (see comment above).
Refer to answer to comment on P6, line 5 ("heated to 40 °C")

**P10, line 1. "This type of treated cylinder exhibits very good long-term stability for a number of VOCs (Gary Barone et al. Restek Corporation, 2010)."** I don't understand this reference, and it is not listed in the Reference section.
The Barone citation referred to a patent for the SilcoNert treatment. It has now been replaced by two references (Allen et al., 2018, and Rhoderick et al., 2019) that studied the stability of VOCs in Silconert treated stainless steel cylinders (amongst other types).

**P10, line 2. "The exact isoprene amount fraction in the secondary standard is determined by validating it against the NPL primary standard."** Can the authors make a statement about the stability of their secondary standards over time? This reviewer has found that secondary isoprene standards made in Aculife [Air Liquide] cylinders can degrade over the timespan of months.
We routinely measure the secondary standards against the NPL primary standard before and after field deployments and we found no statistically significant degradation over the time span of a relatively long field deployment (~ 5 months).
The consensus seems to be to either use Air Products Experis cylinders or SilcoNert2000-treated stainless steel cylinders for the best VOC stability over time (see Allen et al., 2018, and Rhoderick et al., 2019). We have added the following sentence:
"We routinely measure the secondary standards against the primary standard before and after field deployments to account for any degradation over time. However we have found no statistically significant degradation over the time span field deployments (up to 5 months)".

**P10, line 5. "mixing rations"** should be mixing ratios
Amended.

**P10, lines 10-20. "Calibration frequency is specified by the user . . ."** While the explicit description of the instrument calibration method is welcome, it is probably more appropriate to move this material, along with Figure 6, to a supplemental materials section.
We have edited this paragraph to only include relevant information on the calibration process. We feel figure 6 is useful in illustrating how the calibration areas span the dynamic range of the sample areas and therefore wish to retain it in the main body of the manuscript.

**P10, line 26. "a random mixture of 3, 6, 12, 24 and 48 mL samples"** It appears that the calibration sample volumes used are always significantly smaller than the ambient air sample volumes. Since breakthrough volume is a critical parameter of the sample trap (Section 5.1 of text), I am curious as to why the calibration does not include a sample volume larger than ambient volume.
We typically have standards at higher concentrations than ambient isoprene (5-10 ppb), meaning that a smaller volume of the calibration gas is needed to obtain a peak of area comparable in magnitude to that of a sample run. We agree with the reviewer that avoiding breakthrough is critical, so we are looking to introduce periodic larger calibration volumes in the measurement routine to ensure linearity and check for breakthrough.

**P10, line 30. "The equation for the quadratic fit allows the determination of the fractional isoprene amount in the samples by extrapolation or interpolation"** The use of a quadratic calibration curve with extrapolation seems especially susceptible to the introduction of additional error in reporting mixing ratios.
We have now amended all fits to linear as opposed to quadratic.

**P11, line 1. Figure 6.** The linear fits of the peak areas for the individual calibration volumes versus time do not seem appropriate for this figure, since they are not used in the calibration procedure described in the text (peak area versus calibration sample volume on a weekly time scale) nor discussed in the text. The peak areas of 48mL calibration samples show a significant decay with time that is not apparent in the smaller calibration volumes. Is this statistically significant, and, if so, does this imply breakthrough for large sample volumes?
We have replotted Figure 6 with the linear fits removed. We also realised that the subset of 48 mL peak areas shown in the earlier version exaggerated the negative trend, and have included values at earlier dates. This shows a much less pronounced negative gradient for the 48 mL data, much more in line with that of the other calibration volumes and with the overall trend in the instrument response that we attribute to trap degradation (as shown in Figure 16). Linear fits to the new version of Figure 6 (not shown) indicate that the 48 mL areas decrease by ~6.4% over the timescale shown, comparable to the decrease seen at 12 mL (6.9 %). Breakthrough would imply some degree of curvature at higher volumes, and we found curvature to be statistically insignificant for all calibration plots.

**P11, line 9. "The x-axis ('Effective Calibration Concentration') consists in the calibration volume (in mL) multiplied by the isoprene concentration in the gas standard (in ppb)."** Isn't this simply calibrant mass or moles, then? Are you attempting to convert these calibration points to equivalent mixing ratio in ambient air? After solving for the 'Effective Calibration Concentration' based upon ambient peak area, does the user then divide this 'Effective Calibration Concentration' by the sample volume to determine the ambient mixing ratio? The authors should provide the equation for converting peak area and sample volume to mixing ratio.
We have now changed the x-axis for Figure 7 to "number of calibrant moles". We also added the following text to explain how the ambient mixing ratios are calculated from the measured peak areas:
Using the sample peak area ($A_{sam}$), the sample volume ($V_{sam}$) and the intercept ($c$) and gradient ($m$) of the calibration curve, the isoprene mixing ratio in the sample can be calculated using Eq. (2):

$$\chi_{sam} = (A_{sam} - c)/m \cdot (V_{mol}/V_{sam}) \qquad (2)$$

**P11, line 15. "The precision of the instrument was determined as the relative standard deviation in isoprene peak area from calibration chromatograms with the same user-specified volume . . ."** In light of the temporal trend in sensitivity discussed later in the text (5.3.2), does this overestimate the precision uncertainty? The authors do not define the length of time that is averaged here, but it appears that the instrument sensitivity decreases on the time-scale of weeks, and therefore we would not expect the peak area here to be a constant.
We have added a sentence at the end of the preceding Section (Section 4.1) to clarify that the data is analysed in weekly segments, so that a calibration curve is obtained for each week of measurements. This is then echoed in Section 4.2, where we add that the precision is also

calculated on a weekly basis. However upon reanalysis of the precision from the WIsDOM campaign, we have revised down the quoted figure (10.4 from 11.3%) as the average precision from the weekly precision estimates, shown in the Figure below.

[Figure]

**P12, line 4. "The accuracy of the instrument is dictated primarily by the uncertainty in the isoprene amount fraction in the NPL standard, and how this is propagated to the isoprene amount fraction in the secondary gas standard used in the field."** Because the authors have not fully demonstrated that this technique is adequately selective to isoprene, I don't think this statement is correct. Certainty the uncertainty of the calibrant mixing ratio is a component of accuracy, but perhaps not the primary driver. The intercomparison of two iDirac instruments presented in 5.1 would indicate this as well.

Text amended to say: "One of the main components of the accuracy of the instrument is the uncertainty in the isoprene amount fraction in the NPL standard, and how this is propagated to the isoprene amount fraction in the secondary gas standard used in the field."

Additional text added to estimate the impact o long-lived co-eluters on the accuracy as per the answer to point 2 on page 2.

**P12, line 9. "XLGENLINE, a generalised least-squares (GLS) software package for low-degree polynomial fitting (Smith, 2010) is used to estimate the final uncertainty in the isoprene amount fraction in the secondary calibration cylinder by inverse regression from the calibration curve. For most secondary calibration cylinders, this is estimated to ~ 7% at the k = 2 level (providing a coverage probability of approximately 95%). A similar procedure is applied to calibration and sample data from the field to estimate the uncertainty in the ambient isoprene concentrations. This is estimated to ~20-25 % at the k = 2 level."** I found this discussion to be difficult to assess, and reads as simply reporting the output of a statistical software package. Can the authors put this into more explicit terms, e.g. describe what is meant by "inverse regression from the calibration curve"? Also, the comment "most calibration cylinders" is vague – how often do secondary cylinders fall outside this accuracy range? It's not clear how this technique is extended to the field data; again, please describe the analysis in detail. Finally, it should be noted that the authors here use a 2-σ uncertainty to define accuracy, but a 1-σ uncertainty to define precision above. One uncertainty level should be used for both.

We have rephrased the paragraph to clarify the procedure. For further details we recommend the reference cited in the text. We have also amended the coverage factors from 2 to 1 sigma.

**P12, line 15. Figure 8.** The test cylinder mixing ratio has a negative mixing ratio at the start of the experiment shown. Is this real data, or just an artefact of the analysis? Perhaps the data plot should start after the first standard peak area, at 21:00, 21-Feb-2018.
Those data points are an artefact of the analysis, as the instrument is warming up and we are effectively integrating the baseline. The Figure has been amended.

**P12, line 19. ". . . the high concentration of isoprene would risk *poisoning* the adsorption trap."** Italic emphasis is mine. Did you intend to use the word "poisoning" here? I interpret that this to mean an irreversible change to the adsorptive strength of the trap, while I believe that you mean that the trap would demonstrate breakthrough due to non-ideal behavior, as described by Peters and Bakkeren (1994).
We have amended the text as follows:
"… a smaller volume should be used to avoid non-ideal behaviour of the adsorbent as described by Peters and Bakkeren (1994)".

**P12, line 31. "This is identified as the limit of detection and is calculated for two versions of the iDirac, the grey and the grange instruments."** I found this discussion a bit hard to follow since results from these two instruments were presented without introduction. A sentence or two from the later discussion of these instruments in Section 5.1 would be helpful here. Alternatively, simply cite the LOD for the grey instrument here and discuss the higher LOD observed for the orange instrument in section 5.1.
"grange" changed to "orange". We have added a reference to Section 5.1 following the introduction of the two instruments.

**P12, line 32. "This difference is attributed to the traps used (i.e., a trap with more adsorbent would retain more analyte, resulting in a larger signal)"** Doesn't this imply that the traps are not quantitative? If the amount of analyte retained is proportional to the adsorbent mass, this means the traps have a consistent breakthrough, rather than a critical sample volume as described by Peters and Bakkeran (1994). Data provided in the next section (e.g. Figure 11) shows this not to be the case. This explanation does not seem reasonable.
We have amended the text as follows:
"The limit of detection for two versions of the iDirac, the grey and the orange instruments (see Section 5.1 for details), during their deployment in Wytham Woods (See Section 5.2) were 108 ppt and 38.1 ppt respectively. These are higher than the limit of detection determined in the laboratory (46 ppt and 19 ppt respectively). The difference between field and laboratory sensitivity is due to greater noise in the field measurements, as a result of less controlled environmental conditions. The difference in the limit of the detection between the two instruments is attributed to differences in instrumental noise (the noise in the Orange iDirac is 10-20% greater than that from the Grey iDirac), different responses of the PIDs to isoprene, and using traps at different stages of their life cycle (refer to Section 5.3.2 and Figure 16)."

**P13, line 18. "This under-reading is likely due to differences in the absorbent trap, leading to a lower sensitivity for the orange instrument. This is supported by the calibration curve for the orange iDirac, which curves more at high concentrations, resulting in lower peak height than in the grey iDirac for the same concentration.**

**Another artefact of this is that the noise visible on the orange output is greater.”** I am not persuaded by this argument. Why wouldn't calibration account for these differences, irrespective of the curvature of the calibration? Applying a straight-edge to the scatterplot of the two instrument responses shown in Figure 10, I see no significant curvature, i.e. the 6.6% difference appears consistent across the mixing ratio range. See also previous comment for my concern that the adsorbent trap mass is used to explain accuracy differences. The authors may wish to consider alternative explanations, e.g. if the pressures during calibration are the same in both instruments (see comment for P7, line 17). There are two qualitative statements presented here that should be quantified: “curves more at high concentrations” and “noise visible on the orange output is greater.” Finally, it should be noted that the two instruments perform within the specified accuracy for this system (7%), and therefore the performance demonstrated here is acceptable.

Upon closer inspection of the individual chromatograms, we found that the isoprene peaks in the runs from the Orange iDirac exhibited some degree of tailing, particularly evident in the runs where isoprene was higher than 7 ppb. By fitting exponentially modified Gaussians to a subset of these high concentration runs, we found that a normal Gaussians fit underestimated the peak area by approximately 2%. We believe this accounts for some of the observed discrepancy. We have amended the text as follows:

“The results from this experiment are shown in Figure 9. The orange iDirac under-reads by 6.6% relative to the grey iDirac, and this is particularly evident at high concentrations (> 7 ppb). Figure 10 shows this data as a scatter plot of the 15-minute average values from either instrument, again it can be seen that the orange iDirac under-reads slightly. This under-reading is partly attributed to the systematic underestimation of the peak areas from the Orange runs due to peak tailing. Integration of a subset of chromatograms using an exponentially modified Gaussian function showed that a simple Gaussian fit underestimates peak areas from the Orange instrument by up to 2 %. No significant degree of tailing was observed in the runs from the Grey instrument. Despite this slight discrepancy between the output isoprene concentration from the two instruments, the two iDiracs perform within their specified accuracies (see Section 4.2).”

We have also added a statement about peak tailing in the section about accuracy.

We do not believe discussion of the noise in the two instruments is appropriate here, and we have addressed this in the section on sensitivity (Section 4.3; also see answer to the previous comment)

**P14, line 4. Breakthrough tests.** The results presented in this section present a very nice demonstration of the adsorbent trap showing ideal behaviour with a (presumed) single-component isoprene mixture. The citation for this experiment [Peters and Bakkeren, 1994] observed that the breakthrough volume observed in this sort of experiment will be higher than when using a test mixture that includes other compounds, which “may have a pronounced influence on the BTV and thereby on the safe sampling volume of single compounds.” The authors should specify if the test mixture included other species expected in ambient air. If it did not, they should consider how valid the results presented here will be for collection of ambient air.

These tests were carried out using a mixture of 5 ppb isoprene and 5 ppb α-pinene in a nitrogen balance. We have added this detail to the text.

**P14, line 7. “Each run sampled 10 mL”** Did you mean this? Since you specify varying sample volume in figure 11, I assume this to be a typo and should read “sampled 10 mL min-1 .”

The text is correct, but the axis label is unclear. It now reads "Cumulative Sample Volume", indicating this axis is the addition of each individual 10 mL sample run.

**P14, line 8. "isoprene mixture of known concentration"** Perhaps better to say "isoprene mixture at constant concentration."

Amended to state the actual composition of the mixture (see answer to comment P14, line 4. Breakthrough tests above).

**P14, line 16. Co-elution of interfering species.** I found this section to be a bit trivial, due to the somewhat arbitrary list of species used for this test. I wonder if many of the alkenes used for this work (e.g. 2-methyl-1-pentene) would be expected to be found in ambient air at significant mixing ratios to be of importance as an interferent. The authors note "Work is ongoing to determine the elution time of a wider range of compounds, including oxygenated products from the oxidation of isoprene." Since the PID is sensitive to many species beyond alkenes (e.g. ketones, aldehydes), I'm not sure if the work here is very conclusive in demonstrating that there would be no interference when measuring ambient air. I would suggest moving this section to a supplemental materials section, or revising with a more exhaustive list of species.

From an analytical perspective alone, the choice of interfering species tested for in this section is not arbitrary at all, as these are the compounds with elution time closest to isoprene. Since the manuscript was first submitted for review, we have tested for acetone and ethanol and found they did not elute in the chromatographic window used. This rules out two common oxygenated VOCs. In addition, the PID is not sensitive to methanol or formaldehyde, thus narrowing down the list of potential interfering species. We have added the following text to the section, which we think makes more relevant to the ambient measurements described in the following pages:

"Similar tests were carried out for acetone and ethanol, and we found they eluted outside of the chromatographic window considered here."

We have also added a paragraph on the use of peak fit parameters (width, RMSE) to rule out co-elution, in response to point 2 on page 2.

**P15, line 23. "This is an artefact of the trap adsorption process and the resulting tailing of the peak."** This is a nice explanation of the observed phenomenon. Have the authors considered if the tailing for peaks in ambient air relative to peaks from calibration runs leads to a bias in the peak areas? That is, a Gaussian fit function may miss some of the peak tail for ambient runs relative to calibrations, leading to consistent under-reporting of ambient mixing ratio. This effect could be tested by integrating a subset of peaks to capture all peak area, or by using an exponentially-modified Gaussian fit function.

We have tested a subset of sample chromatograms over a range of concentrations, and found that integrating an exponentially-modified Gaussian as opposed to a simple Gaussian function only leads to an increase in area of ~2%.

**P16, line 11. "Sabah (Malaysian Borneo)"** Could the authors be a bit more specific for this location? Sabah is roughly the same size as Scotland.

Text amended to:

"… the iDirac had its first field deployment at the Bukit Atur Global Atmospheric Watch (GAW) station in the Danum Valley Conservation Area in Sabah (Malaysian Borneo)."

**P17, line 7. "Error! Reference source not found."** I believe this is a reference to Figure 13.
Amended.

**P17, line 15. Figure 14.** For the time series shown, there is a significant time period (16-Nov-2015, 19:00 - 2200) where the mixing ratio trace is significantly below zero. Is this correct or a plotting error? Is this attributable to the "several issues with instrument function"? If the latter, perhaps this time period should be referenced in the text.
There were actually some issues with baseline shifts in some chromatograms. These should have been vetted at the inspection stage, and have been removed.

**P18, line 12. "5.23.2 Results and discussion"** Should read 5.3.2 Results and discussion.
Amended.

**P18, line 17. Figure 15.** There is a drop in the mid-canopy time series on the morning of 7-Nov-2018 that seems anomalous with the other mixing ratio traces and other days shown. The authors may want to confirm that those data pass quality assurance checks.
There was an error in the data plotted. We have now plotted the correct data.

**P19, line 2. Figure 16.** The calibration curves presented in this figure give me particular concern, as there appear to be consecutive weeks where there is a positive and then negative curvature. Does this seem physically possible? Generally, I found this figure difficult to interpret as I am color-blind and the color scale to distinguish weeks is quite subtle to me. If the underlying data were re-fit with a linear calibration function and a time-series of the slope were presented instead, I suspect that this figure would be much more effective. The x-axis "Effective Calibration Concentration" should have units of mass or moles, as noted for Figure 7.
The range on the x-axis in the original version of the figure actually exaggerated the curvature of these fits, as the lines were extrapolated well beyond the highest volume (or "Effective Calibration Concentration") on the calibration plot (typically around 600 "Effective concentration" units, *vs* the extrapolation to 1000 "Effective concentration" units in the original Figure 16). As described in the response to the first comment on page 1, we have concluded that the curvature is not statistically significant and have replaced the quadratic fits in Figure 16 with linear fits. We have also added a second panel to the figure with a time series of the gradient of the calibration plot. We have also amended the colour palette to make the figure easier to interpret.

**P19, line 3. "This drift is attributed to the gradual degradation of the trap"** How do the authors separate loss of trapping efficiency from loss of detector sensitivity? If a new sample trap returned sensitivity to the Week 1 level, that would be a useful data point to show.
We have added an extra point to the plot at week 20, showing the calibration line and its gradient after replacing the trap. This shows an enhanced response, confirming that loss of trapping efficiency (and not detector drift) was the cause of the observed drift.

**P19, line 26. "a more sophisticated and interactive control over the oven temperature."** This statement begs for a presentation of underlying data (oven temperature time trace, retention time drift) to justify this need. Possibly, either could be added to field data time traces in Figure 6, 14 and/or 15 on a secondary axis.

In response to "P6, line 5", we have added a more in-depth description of the temperature variation and how we deal with its effect on elution times.

**P20, line 9. References**. At least two references (Allen et al., 2018; Helmig et al., 1998) are incomplete. The Barone et al., 2010 reference from the text is missing; see previous comment for P10, line 1.

The Barone reference (a patent) has been replaced with the Allen et al. (2018) reference and an additional reference (Rhoderick et al., 2019), which are the correct works to support that statement. The Allen and Helmig references have been amended.

**References**

- Barket et al., J. Geophys. Res., 106: D20, 24301-24313 (2001), doi.org/10.1029/2000JD900562
- Fu, D. et al., Nature Communications, 10: 3811 (2019), doi.org/10.1038/s41467-019-11835-0
- Peters and Bakkeren, Analyst, 119, 71-74 (1994), doi.org/10.1039/AN9941900071

---

## Author Comment (AC2) · 3 Jan 2020

Reviewers' comments are in black, authors' response in blue and changes to the text in red.

**Reviewer #2**

This paper thoroughly details the trial of a deployable instrument for remotely measuring isoprene (the main focus of the manuscript) and similar VOCs. There is a great need for calibrated portable instruments that can monitor long term trends out in the field for campaigns and background measurements therefore the iDirac instrument is of great interest, as this can complement reference materials and satellite measurements to give a holistic overview of the atmospheric isoprene distribution. The paper thoroughly introduces the instrumentation and shows the breadth of measurements that have been performed to test it. However, despite the wealth of information specific quantification is often not provided (please see Specific comments). I recommend that this manuscript be accepted once these minor corrections have been addressed.

We thank the reviewer for their feedback and address their comments below.

**General comments**

In the introduction it would be useful to explain the importance of isoprene with regards to impact on the OH reservoir in the troposphere, as well as on SOA.
We added the following:
"As a result of its reactivity and the magnitude of its emission rates, determining the global abundance of isoprene is important to understand the oxidising capacity of the atmosphere (Squire et al., 2015) and the formation of SOA, which can affect the optical properties of the atmosphere and in turn impact the climate (Carslaw et al., 2010)."

I think that a bit more detail is required about the instrumentation, particularly with regards to the drier and breakthrough and poisoning of the trap.

Have any tests been performed to assess if the drier removes any VOCs?
Tests were carried out with and without the drier while feeding the instrument the same mixture. No statistically significant differences were observed. We added this sentence to the text:
"Laboratory tests found no statistically significant difference in isoprene peak area between runs using the drier and runs bypassing it."

How do you recondition the trap, what if terpenes or heavy components stick and reduce trapping ability?
We run blank runs periodically to make sure heavier components are not carried over between runs. However we have not found evidence for carryover. We monitor the trap performance using plots such as the one shown in Fig. 16 and eventually replace it with a new one.

Figure 9
– breakthrough affected by flow rate
– how were the samples pumped into the system
– what happens if you vary the sampling rate?

Upon closer inspection, we now attribute the difference between the two instruments to peak tailing in the chromatograms from the Orange instrument (leading to an underestimate of the peak areas). We have amended this whole section to reflect this.

Similar questions arise from the results depicted in Figure 11.
– breakthrough affected by flow rate
The flow rate is held constant (~ 20 mL/min, the same flow used in the field) so the breakthrough volume obtained here is relevant to our field measurements.
– how were the samples pumped into the system
This is as described in Section 2.2
– what happens if you vary the sampling rate?
We have only investigated this using the sampling flow rate used routinely in the field. An in-depth investigation on the effect of flow rate on breakthrough volume would be valuable, but it is beyond the scope of this work.

It would also be helpful to have a clearer idea of what is meant by the trap becoming "poisoned" is this the result of moisture and what is the impact on uncertainties and sensitivity?
We believe the adsorbent degradation after repeated heat cycles is the principal cause of trap degradation. Typically thermal desorption tubes are replaced after 100 heat cycles. A week of continuous operation of the iDirac in the field consists of ~1000 heat cycles. We have added a second panel to Figure 16 to show the decrease in sensitivity as a function of time.

I strongly suggest that Figure 3 includes information about the isoprene concentration depicted and I would also like error bars added to all figures where required e.g. Figure 7.
Figure 3 has been amended to include a calibration, a sample and a blank chromatogram with peak fits. We also added additional information in the caption regarding the isoprene concentration for each run.
Error bars were added to Figure 7.

Volumes should have an associated uncertainty.
Refer to the answer to comment P10. Line 26.

Finally, I think it would be useful to know why was a Gaussian shape was used for the fitting, have you tried any other peak shapes for fitting e.g. Voigt or speed-dependent Voigt? These might optimise results.
We found that Gaussian curves provided the best fit to the observed chromatographic peaks. We found a marginal improvement with exponentially-modified Gaussians curves.

Specific comments
P1. Line 23. Please change to "Isoprene is an important nonmethane"
We prefer to stress the importance of isoprene relative to other non-methane VOCs, so we have not changed the text.

P1. Line 25. What is the impact of the SOA, please also add references
We added the following:
"As a result of its reactivity and the magnitude of its emission rates, determining the global abundance of isoprene is important to understand the oxidising capacity of the atmosphere

(Squire et al., 2015) and the formation of SOA, which can affect the optical properties of the atmosphere and in turn impact the climate (Carslaw et al., 2010)."

P2. Line 16. Grab samplers: please add example reference e.g. Robinson AD DOI: 10.5194/acp-5-1423-2005
Reference added.

P2. Line 36. What about trueness?
"Accuracy" includes trueness and precision according to ISO 5725.

P3. Table 1. Please specify nitrogen purity percentage
Text amended as follows:
High Purity Nitrogen (Grade 5.0, or 99.999%)

P4. Figure 1. Does the packaging and foam emit any VOCs?
Not to our knowledge. However the gas lines are never exposed to the internal packaging during a routine measurement sequence, so chances of VOCs from the foam or packaging affecting the measurement are very low.

P5. Paragraph 2. What is the volume sampled?
Details on this are given in the "Sample adsorption/desorption system" section further down P5.

P5. Line 30. Why can you not use a non-return valve?
We have more experience in the use of flow restrictors.

P5. Line 43. Specify the desired volume
This is specified by the user, and we've added the following to this:
"When the desire volume (as specified by the user in the configuration step – see Section 3) is reached…"

P6. Lines 11-14. Consider combining sentences.
We believe the paragraph reads better as two shorter sentences.

P6. Line 21. Are there any other VOCs with similar ionisation values or boiling points?
Most VOCs have ionisation energies in the 8-10 eV range, so they can be detected by the PID (notable exceptions: methanol and formaldehyde). The boiling point is key to determine elution order and potential co-elution, and we have tested for these potential interfering species in Section 5.1.

P6. Figure 3. What concentration does this peak represent? What's the S/N ratio?
We have now amended Figure 3 to include a blank, a calibration and a sample peak, with additional information on the gas sampled in the cation. The S/N ratio for the calibration run is 51.8.

P7. Line 7. Should read "in a nitrogen balance"
Amended.

P7. Line 9. Is the calibration gas purchased or decanted, if the latter please specify how.
As stated in the text, these details are given in Section 4.1 and the reader is referred to that Section for further information on the calibration routine.

P7. Line 20. Please specify the reduced pressure
This is stated on line 23 on the same page (20 kPa below ambient).

P7. Line 23. I think that you mean "nominally" not "typically"?
Agreed and amended.

P7. Line 38. Is the clock calibrated?
The clock is not calibrated but we found negligible drift (1-2 seconds) over a 5 months field deployment.

P9. Line 15. What are the criteria for insufficient/sufficient (how many?)
P9. Line 16. What is the criteria for "too great"
In order to capture the fluctuations in elution time brought about by the variations in oven temperature, we recommend having a calibration run at least every six hours. We have amended the text as follows:
"When there are insufficient calibration chromatograms to determine the isoprene peak retention time (e.g., less than 4 calibration runs in a day)…"

P9. Line 26. What is the stability period of the gas standard?
Studies (Rhoderick et al., 2019)have shown that VOC mixtures in Air Products Experis cylinders have a stability of at least 2 years. We have added this to the text.

P10. Line 13. Replace "good practice" with "essential"!
Amended.

P10. Line 14-19. Please give numbers.
The following text was added:
Typically, a calibration run is performed every 35 sample runs.  As the mean duration of a 150 mL sample run is approximately 9 min (consisting of 7.5 min of sampling and 1.5 min of chromatographic run), a calibration run is performed approximately every 5.25 hours.

P10. Line 26. What is the error on the volumes?
We have added the following sentence (and error bars in the relevant plots):
The error in the sampled volumes is dominated by the dead volume in the gas lines before the trap (approximately 1.6 mL), combined with the uncertainty in the measurement of flow rates (1%) and sampling times (0.05%). The overall uncertainty in the volumes is estimated as 50% for 3 mL, 13% for 12 mL, 3% for 48 mL and 1% for 200 mL.

P11. Paragraph 1. Are we talking about intermediate precision or reproducibility?
This is over the timescale of 1 week to 1 month, so strictly speaking it is intermediate precision.

P12. Line 13. What is the lowest volume used – this may impact uncertainty and S/N as the sensitivity is likely to vary with volume size
The volume in the sample runs is held constant throughout a deployment.

P12. Line 31. "the grey and the grange instruments" I think you mean "orange"; please clarify (explain what the difference is before section 5.5).
"Grange" was a typo and has been amended. We have added a reference to Section 5.1 where the two instruments are described more in detail.

P13. Line 11. Specify uncertainty of BOC mixture and add the word "balance" to describe the matrix gas.
Amended.

P13. Line 12. Do you have an offset from losses to the chamber wall?
Potentially, but the point is that both instruments were sampling the same air. So if there were wall losses within the chamber, they would affect both instruments

P13. Line 15. Should be "tee-piece"
Amended.

P13. Line 17. "high" Specify above 8 ppb.
Amended.

P13. Line 22. What happens if you switched the trap? Have you considered the impact of breakthrough at high flow rates?
The rate of gas flows through the trap is roughly constant (20 mL/min ±2%),
Switching the traps between the 2 instruments would be an interesting experiment, and would definitely allow to quantify the impact of the traps on the overall instrument performance. However we now attribute the difference between the two instruments to peak tailing in the chromatograms from the Orange instrument (leading to an underestimate of the peak areas). We have amended this whole section of the manuscript to reflect this.

P15. Line 8. Rephrase "pure substance". Surely there will be impurities in the raw materials? Perhaps state the purity of the reagent?
Rephrased as follows:
"… from the "pure" substance (Sigma Aldrich, purity typically > 98%)…"

P16. Figure 12. What does a blank run look like?
A typical blank run was added to Figure 3.

P17. Line 7. Please resolve "Error! Reference source not found: : :"
Amended.

P17. Line 12. How has this been addressed in subsequent versions?
We implemented an "idle" setting for when the instrument is not used in between measurements. This way, once warmed up, the iDirac can be left on stand-by without requiring another warm-up period when measurement is resumed.

P19. Figure 16. Can you please use a wider colour range?
We have amended the colour palette.

P19. Line 4. Is the poisoning moisture? What impact does this have on the sensitivity and uncertainties attributed?

We believe the adsorbent degradation after repeated heat cycles is the principal cause of trap degradation. Typically thermal desorption tubes are replaced after 100 heat cycles. A week of continuous operation of the iDirac in the field consists of ~1000 heat cycles. We have added a second panel to Figure 16 to show the decrease in sensitivity as a function of time.

P19. Line 20. "can be run autonomously for months" Assuming the trap is not degraded?

Even as the trap degrades progressively, the instrument can run on the same trap for up to 19 weeks as shown in Figure 16. We have added the following sentence to the text:

"provided the performance of the trap is assessed periodically"